# communications
# earth & environment

# Climate change is narrowing and shifting prescribed fire windows in western United States

Daniel L. Swain [1,2,3✉], John T. Abatzoglou [4], Crystal Kolden[4], Kristen Shive[3,5], Dmitri A. Kalashnikov[6], Deepti Singh [6] & Edward Smith[3]

Escalating wildfire activity in the western United States has accelerated adverse societal impacts. Observed increases in wildfire severity and impacts to communities have diverse anthropogenic causes—including the legacy of fire suppression policies, increased development in high-risk zones, and aridification by a warming climate. However, the intentional use of fire as a vegetation management tool, known as "prescribed fire," can reduce the risk of destructive fires and restore ecosystem resilience. Prescribed fire implementation is subject to multiple constraints, including the number of days characterized by weather and vegetation conditions conducive to achieving desired outcomes. Here, we quantify observed and projected trends in the frequency and seasonality of western United States prescribed fire days. We find that while ~2 C of global warming by 2060 will reduce such days overall (−17%), particularly during spring (−25%) and summer (−31%), winter (+4%) may increasingly emerge as a comparatively favorable window for prescribed fire especially in northern states.

[1] Institute of the Environment and Sustainability, University of California, Los Angeles, Los Angeles, CA, USA. [2] Capacity Center for Climate and Weather Extremes, National Center for Atmospheric Research, Boulder, CO, USA. [3] The Nature Conservancy of California, Sacramento, CA, USA. [4] Management of Complex Systems Department, University of California, Merced, Merced, CA, USA. [5] Environmental Science, Policy and Management Department, University of California, Berkeley, Berkeley, CA, USA. [6] School of the Environment, Washington State University, Vancouver, WA, USA. ✉email: dlswain@ucla.edu

The escalating wildfire crisis across much of the American West has garnered considerable international attention due to increasingly severe societal and ecological outcomes. Such impacts–including both the direct effects of destructive wildfires (thousands of structures destroyed and hundreds of civilians killed[1]) as well as the indirect effects of prolonged air pollution episodes yielding greatly increased excess morbidity and mortality[2–6]) and increased post-fire hydrologic hazards[7]—have spurred urgent policy conversations centered on solutions.

Recent work points toward anthropogenic climate change rapidly altering Western United States (WUS) fire regimes to produce overall more extreme wildfires[8–12] that burn over a longer fire season[13,14], at higher elevations[15], and with greater synchroneity across multiple regions[16,17], ultimately producing more smoke and greater carbon emissions[18]. Although most ecosystems across the WUS are fire-adapted, recent increases in the severity and frequency of wildfires have become disruptive–causing the loss of old-growth forest[19], ecosystem type conversions[20], and reductions in carbon storage[21].

However, even as projections suggest future warming will further amplify these trends (e.g.,[17,22]), other non-climate factors are driving profound changes in wildfire regimes, each with different near-term local solutions[23]. These stressors include excessive fuel accumulation in some ecosystems due to 20th-century fire exclusion[24,25], changing patterns of human-caused ignitions[26], and the expansion of populated areas into high-risk zones[27].

Prominent among proposed strategies in addressing the WUS wildfire challenge is the use of prescribed fire (also known as controlled burning), which is the practice of intentionally igniting and managing fire under prescribed conditions to meet specific desired hazard reduction or ecosystem-related objectives–including the reduction of wildland fuel density and improving ecosystem health and resilience to a warming climate and other disturbances. Prescribed fires in the WUS are generally conducted by fire personnel with state or federal agencies, though private landowners and nonprofit entities are increasingly conducting prescribed fires as well. Cultural burning is a related practice involving the use of fire by Indigenous peoples that includes some goals overlapping those of prescribed fire (management of natural landscapes in a way that ultimately reduces vegetation density and subsequent fuel loading), but that also holds much broader cultural importance and is practiced using a more holistic knowledge of place to guide the timing and implementation of burning activities[28,29]. This manuscript is focused on prescribed fire as practiced by government agencies and other organizations that follow their standards (e.g., The Nature Conservancy), but the issues explored have relevance to cultural burning as well, given the importance of ambient weather and vegetation conditions in both settings.

Historically, most prescribed burning in the WUS has occurred during spring or autumn, when weather and vegetation conditions are more likely to result in fire behavior that meets objectives but can still be controlled. These conditions collectively make up the "burn prescription" —or specified ranges of weather and live and dead vegetation moisture parameters that ensure fires burn completely enough to achieve objectives (e.g., consuming large fuels and woody debris) but that they don't burn so hot as to present control problems or to have undesired ecological consequences (e.g., increased tree mortality). Periods of time where these prescribed conditions are likely to be met are called "burn windows."

Although considered a widely applicable solution, there are many impediments to prescribed fire implementation[30,31], including staffing and funding limitations, risk tolerance, and smoke impacts[32]). For these and other reasons, prescribed fires are not implemented during all suitable burn windows—suggesting that, to date, climate has not been the primary inhibitor to implementation and that present-day burn windows are often underutilized[33–35]. However, in recent years, the combined effects of severe short-term drought and long-term aridification[16] have contributed to a reduction of adequate spring and autumn burn windows in some regions[32,35], raising concerns that climate change will add to the many existing challenges to prescribed fire implementation[36].

Beyond direct meteorological constraints, climate change has already demonstrated indirect impacts on prescribed fire implementation. Extreme meteorological events, especially severe to historically unprecedented drought conditions, have been contributing factors to several prescribed fires that "escaped" and became disastrous wildfires. The societal and political fallout from such events has led to multiple temporary U.S.-wide moratoria on prescribed fire (such as after the 2000 Cerro Grande Fire[37] and the 2022 Hermits Peak Fire[38]. Longer fire seasons also mean that fire personnel are committed to fighting fire in some regions–which in turn reduces the number of trained personnel available to implement burns in other regions where conditions are favorable. In addition, longer fire seasons are particularly challenging for federal agencies since their firefighting workforce is dominated by seasonal employees who can only work for limited durations[33–35].

Given calls for a substantial expansion of prescribed fire implementation to combat wildfire risk[30,39,40], there is a critical need to understand the extent to which climate change alters the seasonality and frequency of burn windows. To significantly increase prescribed fire implementation, the other impediments to prescribed fire will also need to be addressed; however, understanding potential shifts in burn windows could empower state and federal agencies to create more realistic staffing plans that maximize the potential for prescribed fire implementation.

Here, we quantify both historical shifts as well as projected future changes in burn windows favorable for prescribed fire. We find that ~2 °C of global warming will decrease the overall number of days per year conducive to prescribed burning by −1% to −29% (−0.5 to −16.6 days per year, depending on subregion), with most of those decreases occurring in spring and summer. However, we also find increases in days suitable for prescribed burning during winter (+4%, or +0.6 days per year)—potentially increasing opportunities for burning outside of current target seasons, especially across the northern interior WUS. We also explore and highlight the potential importance of accounting for large-diameter fuel moisture (FM) as a proxy for the impact of long-term drying trends on woody fuels (i.e., primarily in forested ecosystems), which to date has only been included in a limited number of WUS burn plans. These large diameter (i.e., 1000-h) fuels have also become increasingly important in a fuels management context following mass tree die-off events due to drought and bark beetles in this region[41,42], as well as in the context of managing smoke-related air quality concerns.

## Results

**Historical seasonality of WUS-prescribed fire windows.** The occurrence of days favorable for prescribed fire from a meteorological and vegetation moisture perspective (henceforth, "RxDays") varies widely across the WUS. Using climate data from 1981 to 2020 in the observed record (GridMET), we calculate the total annual occurrence of RxDays across the WUS (Fig. 1). We define RxDays as days with surface weather (temperature, relative humidity, and wind) and vegetation moisture (1, 10, 100, and 1000-h FM plus ignition component) conditions that are deemed suitable for prescribed fire based on prescriptions drawn

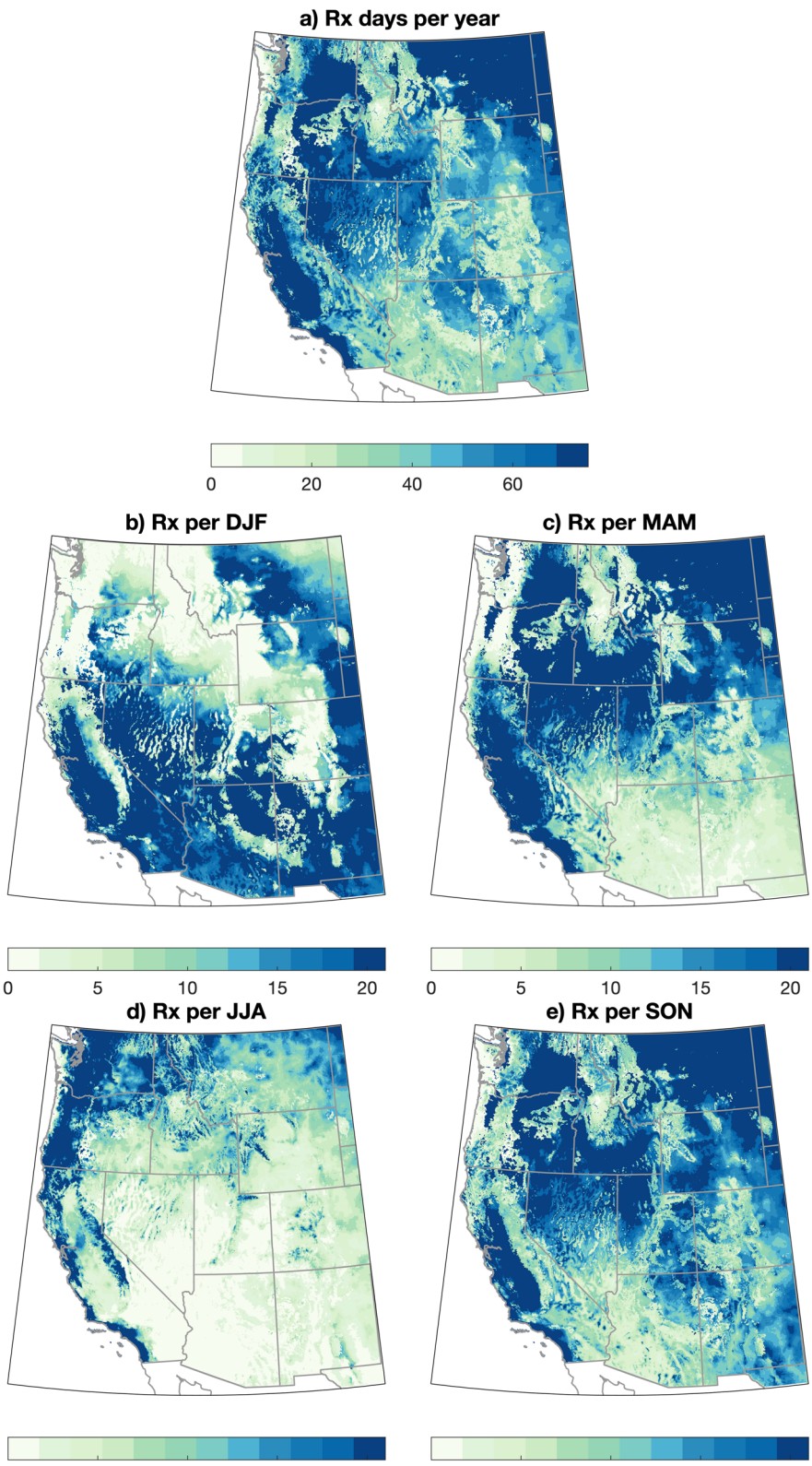

**Fig. 1 Maps of observed RxDay climatology across the WUS.** Maps depicting the observed number of RxDays across the western United States (WUS) on an annual (**a**) and seasonal (**b**–**e**) basis. Observed RxDays are calculated using meteorological data from the gridMET dataset over the years 1981–2020.

from existing real-world burn plans (separately for forested and non-forested ecosystems; see "Methods" and Fig. S1). The background number of annual RxDays ranges from fewer than 10 days per year in extremely moist regions of western

Washington and Oregon to greater than 70 days per year in multiple subregions, mainly those characterized by non-forest vegetation types (Fig. 1a). These bookends reflect the FM constraint in the underlying RxDays definition: in some very dry and

hot regions, vegetation may nearly always be too dry to burn at a sufficiently low intensity, and in some very moist and cool regions, vegetation may nearly always be too damp to burn at sufficiently high intensity (or even at all). Further, a majority of the non-forest covered portions of the WUS (as well as some forested regions) experience at least one cumulative month (≥30 days) of RxDays per year, and many non-forested areas experience two or more cumulative months (≥60 days) of RxDays per year—highlighting the widespread climatic potential for prescribed fire to be utilized across a diverse range of regional sub-climates throughout the WUS (Fig. 1).

Large seasonal variations in RxDay occurrence are also apparent (Fig. 1b–e). RxDays are maximized during winter (DJF) in the southwestern portion of the domain and in summer (JJA) in the northwestern portion of the domain. This overall seasonal progression closely follows the latitudinal cycle of the polar jet stream and associated precipitation-bearing storms over the WUS, which retracts northward during the warm season and southward during the cool season[43]. This seasonal reversal is responsible for a majority of seasonal variation in precipitation across the WUS outside of areas strongly affected by the North American Monsoon, which receives a substantial fraction of their annual precipitation during summer. RxDays in spring (MAM) are concentrated in a latitudinal arc extending from California northeastward into the Northern Rockies, while RxDays in autumn (SON) are more broadly distributed across much of the WUS. Notably, there are multiple sub-regions that experience a substantial number of RxDays in all four calendar seasons.

We also find that most of the WUS exhibits relatively high interannual variability of RxDays (Fig. S2), with a standard deviation of at least ~10–15 days per year in a majority of locations (and locally 20 or more). Such year-to-year variations in prescribed fire window length can significantly affect planning efforts.

**Historical trends in prescribed fire windows**. Observed trends in RxDays vary across the WUS (Fig. 2). Annual RxDays between 1981 and 2020 have generally decreased across the Southwest and increased across the central and northern Rocky Mountains and eastern Washington and Oregon (Fig. 2a). Substantial seasonal differences in observed RxDay trends are also apparent. In DJF (Fig. 2b), strong decreases in RxDays are observed in the interior Southwest and Rocky Mountain Front Range; in JJA (Fig. 2d), RxDay decreases are widespread along the West Coast and across much of California generally, but increases in RxDays are observed east of the Cascades in Washington and Oregon and broadly across the eastern portion of the domain. Transition season (spring and autumn) RxDay trend patterns are more heterogeneous across the WUS (Fig. 2c, e), though MAM somewhat resembles DJF trends (with RxDay decreases across the interior Southwest and increases across the central and northern Rockies, Fig. 2c).

Simulated trends in RxDays over the same historical period (1981–2020) using downscaled climate model data (from a subset of CMIP5 model members in the MACA dataset; see "Methods") are notably different than observations–depicting a more spatially coherent decrease in RxDays over most of the WUS except for a band of increasing RxDays extending from the deserts of south-eastern California eastward to the Four Corners region (Fig. S3). The strongest simulated RxDay decreases are centered broadly across the central and northern portions of the domain, with decreases exceeding 20 RxDays per year in some areas (Fig. S3a). Strong seasonality is present in projected historical trends, with strong DJF increases in the Southwest and weaker increases more broadly in MAM, contrasting widespread RxDay decreases across most of the domains in JJA and SON (Fig S3b–e). The most

spatially coherent region of negative modeled RxDay trends in MAM is across much of Arizona and New Mexico—which is notable given the occurrence of destructive escaped prescribed fires (e.g., the Cerro Grande Fire in 2000 and Calf Canyon/Hermits Peak Fire in 2022) in this region during spring[36].

Observed divergences between observed (GridMET) RxDay trends and climate model-derived (CMIP5) trends have multiple plausible causes. First, observations represent only a single "realization" of all possible sequences of internal climate variability over the 40-year period, whereas simulations of these same years represent the ensemble average of 18 independent models that differ due to both internal climate variability and different physical representations of the climate system. This higher "signal to noise ratio" suggests that the ensemble average of the climate model simulations more likely represents the true multidecadal forced signal (a known advantage of large ensembles, e.g.,[44]). Second, we hypothesize that stronger observed vegetation drying across much of the WUS compared with climate model projections—perhaps due to a combination of natural variability and underestimated regional land surface feedbacks (e.g.,[45])—may have increased RxDays in some regions. This seemingly paradoxical effect would arise from the range of acceptable vegetation moisture in our RxDay definition: not only can ambient conditions be too dry to burn safely, but they can also be too moist to burn with sufficient combustion efficiency to achieve desired outcomes. Thus, in locations where fuels were historically too moist, rapid fuel drying during 1981–2020 decreased FM sufficiently to meet RxDay criteria.

To test this hypothesis, we re-calculate observed 1981–2020 RxDay trends using a definition that excludes all FM constraints (i.e., using only meteorological criteria). Using this restricted definition, we find widespread decreases in RxDays on an annual mean basis (Fig. S4)—with strong decreases occurring nearly everywhere and in all seasons except in DJF (where trends are near zero or slightly positive). This suggests that vegetation aridification in recent decades has indeed yielded a net increase in RxDays in some historically cool and moist areas but that further drying would have the potential to push vegetation in this region "out of prescription" once again if fuels fall below minimum moisture thresholds.

**Future trends in prescribed fire windows**. We use a "medium emissions/moderate warming" greenhouse gas emissions scenario (RCP4.5) to estimate the near-term (2021–2060) effect of climate change on RxDays across the WUS using downscaled climate model data (MACA)[46]. We find that anthropogenic climate change will further decrease RxDays across most of the WUS, with widespread decreases of ~15–30 RxDays per year across the southern half of the domain (including the Pacific Southwest and Four Corners regions) (Fig. 3a). In the northern half of the domain, however, projected trends are weaker, ranging from modest RxDay decreases (5–15 per year) along the northern rocky mountain front range and interior Northwest (east of the Cascades) to little change or even localized increases in RxDays over the Northern Rockies (especially western Montana). We also find strong ensemble agreement regarding both the sign and magnitude of WUS-wide annual changes in RxDays over the 1981–2060 period, with 14 of 18 model ensemble members depicting a decrease of at least 10 RxDays/year (vs. an ensemble mean decrease of ~15 RxDays/year, Fig. S5).

Pronounced seasonal differences in the spatial patterns of projected RxDay changes are apparent. Widespread RxDay decreases are observed across almost the entire WUS domain during MAM and JJA (except portions of the Northwest in MAM). Notably, the strongest JJA RxDay decreases (exceeding 15

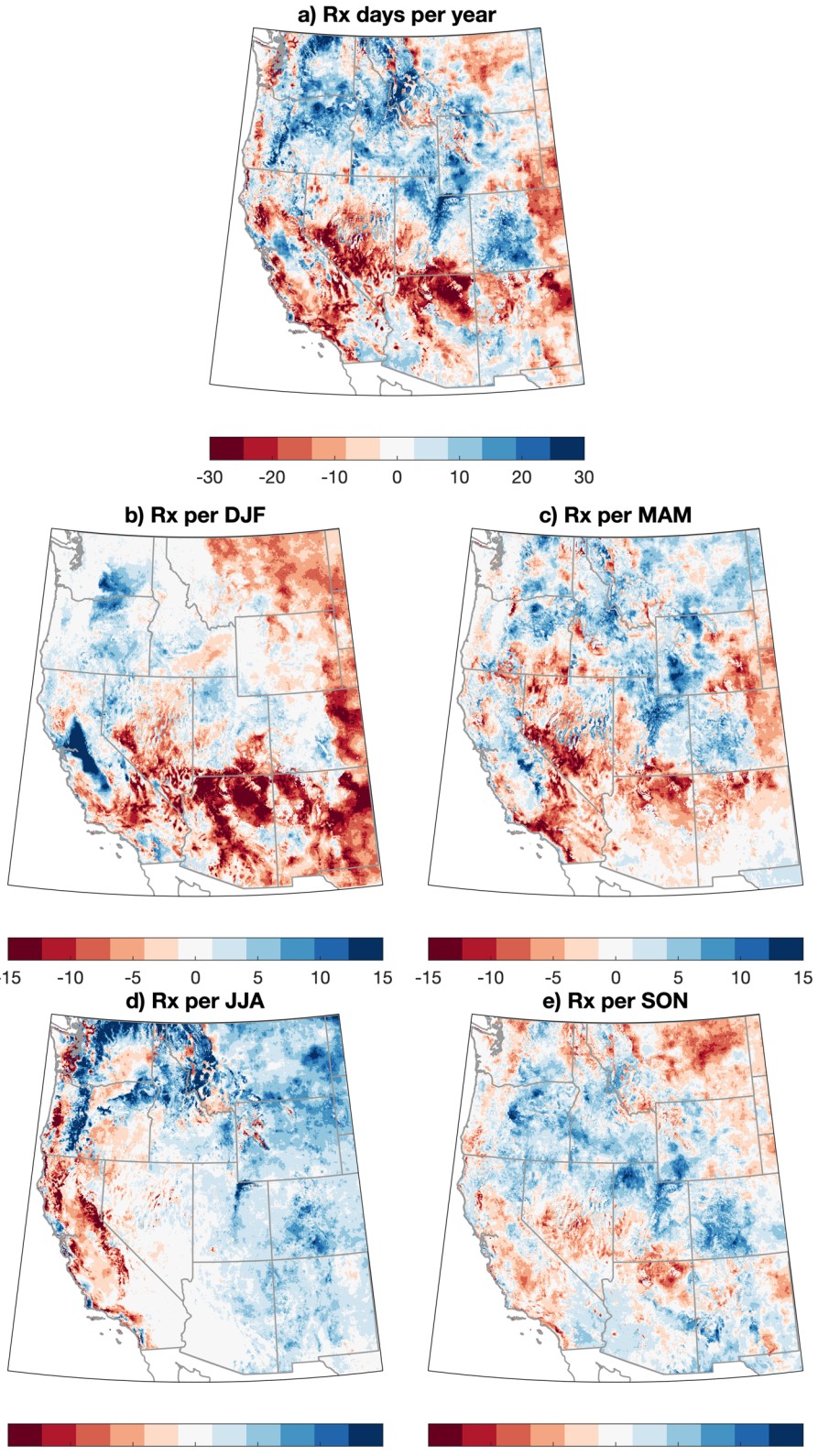

**Fig. 2 Maps of observed trends in RxDays across the WUS, 1981–2020.** Maps depicting the change in the number of observed RxDays across the western United States (WUS) on an annual (**a**) and seasonal (**b**–**e**) basis. The trend in RxDays is calculated using a linear least squares regression and is represented graphically by the accumulated change (in RxDays per year or season) over the full 40-year period using meteorological data from the GridMET dataset (1981–2020).

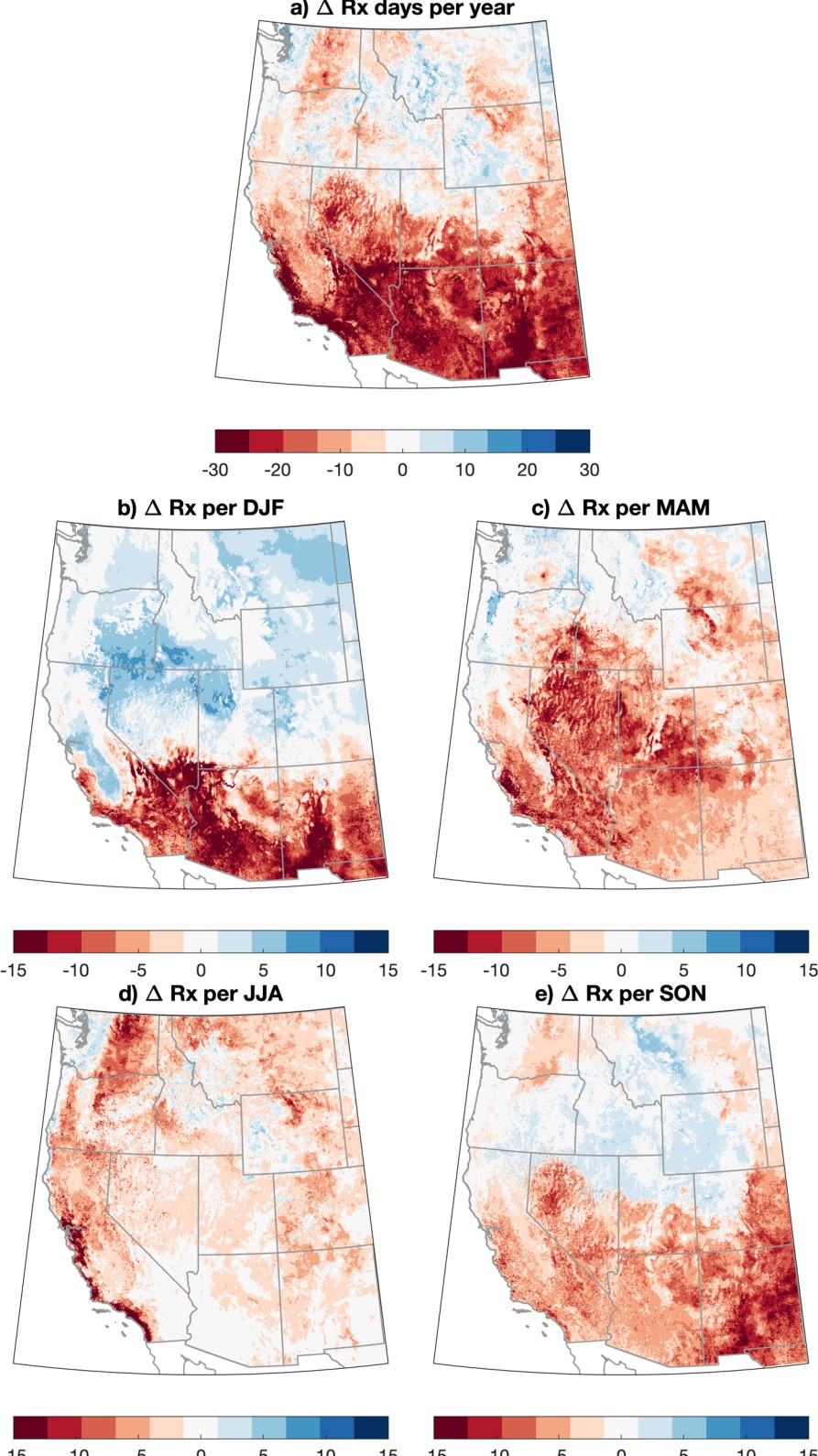

**Fig. 3 Maps of projected change in RxDays across the WUS, 2021–2060 vs. 1981–2020.** Maps depicting the projected change in the number of RxDays across the western United States (WUS) on an annual (**a**) and seasonal (**b**–**e**) basis. The change in projected RxDays is calculated as the difference between the RxDay counts during 2021–2060 vs. 1981–2020 on a "moderate warming" (RCP4.5) trajectory using meteorological data from the downscaled CMIP5 climate model ensemble dataset (MACA).

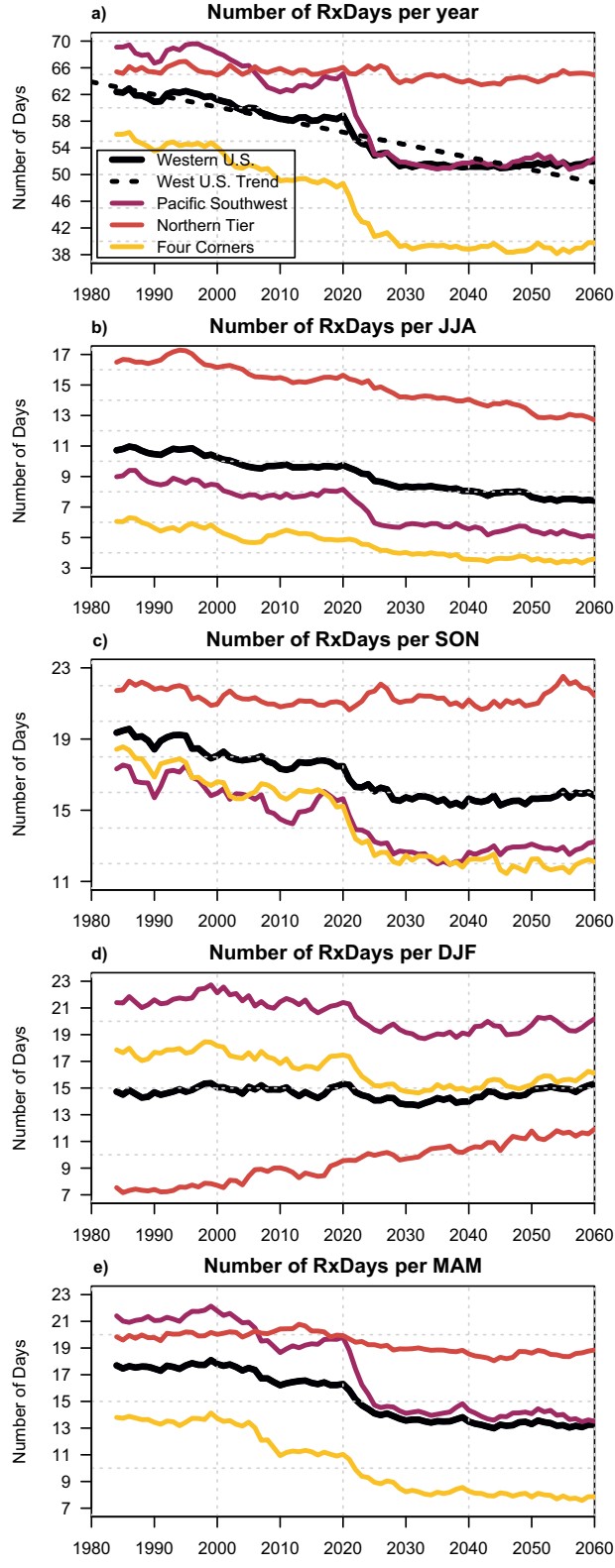

**Fig. 4 Time series of projected trends in RxDays across the WUS, 1981–2060.** Time series depicting the change in the number of observed RxDays across the western United States (WUS) on an annual (**a**) and seasonal (**b**–**e**) basis. Black curves represent the WUS-wide domain average values; magenta, red, and yellow curves represent the sub-regionally averaged values for the Pacific Southwest, Northern Tier, and Four Corners portions subsets, respectively. All plotted data is smoothed on a 5-year running mean basis, and the dashed black line depicts a fitted linear regression using WUS-wide annual average values. All RxDay projections assume a "moderate warming" (RCP4.5) trajectory and use meteorological data from the downscaled CMIP5 climate model ensemble dataset (MACA) over the years 1981–2060.

To systematically contextualize longer-term RxDay trajectories caused by projected climate change, we assess simulated trends in regionally aggregated RxDays over the full 1981–2060 period (by concatenating the Historical and RCP4.5 climate model forcing scenarios). We further define three sub-regions within the broader WUS study domain: the "Pacific Southwest," the "Northern Tier," and the "Four Corners" (see "Methods"). On a WUS-wide annual mean basis, we find a statistically significant decrease in the annual number of RxDays between 1981 and 2060 (−17% or −10.4 RxDays per year, $p < 0.0001$; Fig. 4a). There are strong decreases during JJA (−31% or −3.3 RxDays per season), MAM (−25% or −4.4 RxDays per season), and SON (−22% or −3.5 RxDays per season) (Fig. 4b–d), but a slight increase during DJF (+4% or +0.6 RxDays per season). This modest domain-averaged winter increase is driven primarily by larger RxDay trends across the Northern Tier (+58%, or +4.4 RxDays per season) and neutral or negative trends elsewhere (Fig. 4d). Regionally, these projected decreases are largest in Four Corners (−29%, or −16.2 RxDays per year) and Pacific Southwest (−24%, or −16.6 RxDays per year), and smallest in Northern Tier (−1%, or −0.5 RxDays per year) (Fig. 4a).

Across regions and seasons, the largest increment of simulated RxDay decrease occurs in the first ~50 years (i.e., 1981–2030)—suggesting that the rate of further RxDay decline might slow appreciably in the coming decades on the RCP4.5 trajectory (Fig. 4). However, because observed historical (1981–2020) decreases in RxDays are less uniform and generally weaker than projected by climate models over the same period, it is possible that much of the WUS had not yet (as of 2020) "realized" the expected climate-driven decrease in RxDays that might otherwise have occurred over that period (Fig. 2. vs. Fig. S3). This suggests that the near-term (2021–2060) future trajectory toward fewer RxDays across most of the WUS could potentially be steeper than depicted here.

**Importance of large-diameter FM and long-term drying**. Given recent high-profile public discussions regarding the implications of escaped prescribed fires occurring under conditions characterized by unusually low vegetation and dead FM[37], including specific calls to use indices relevant to large-diameter FM (e.g., energy release component)[36,38], we quantify the importance of considering large-diameter dead FM in WUS prescribed burn plans. To do so, we recalculate our historical RxDay baseline excluding 100-h and 1000-h dead FM constraints (i.e., consideration of large diameter woody debris (~3 cm to 20 cm in diameter), which includes heavy dead brush as well as downed trees and logs, for comparison to our original RxDay definition (which includes these constraints).

Historically, 1000-h FM has not been regularly considered in many burn prescriptions for forested areas in the WUS (of the 14 burn plans implemented in predominantly forested areas used in this study, only one includes a 1000-h FM constraint) because it was a minimal component of broadcast burns where objectives

RxDays per season) are along coastal portions of central and southern California (Fig. 3d). During SON and especially DJF, a trend "dipole" emerges (Fig. 3b, d) wherein strong RxDay decreases are noted across the southern tier (including California, Arizona, and New Mexico) but the rest of the WUS experiences little change or even slight increases in RxDays (especially across the central and northern Rockies and northern Great Basin).

were to consume smaller fuels and large FM was generally too high during burn windows for such fuels to combust. However, with warming temperatures, this historical assumption may no longer be valid as large diameter FM can be used as a proxy for longer-term trends in drought and aridity. Specifically, recent reviews have called for the use of drought indices (such as the evaporative demand drought index) from the U.S. National Fire Danger Rating System (NFDRS) indices in burn prescriptions[36,38], some of which consider large-diameter FM (e.g., energy release component). Recent work has also suggested that larger-diameter fuels can contribute to "mass fire" events[41,47], suggesting they have a greater influence on fire behavior than previously believed and warranting their explicit consideration.

We find, where such fuels occur, that large-diameter FM is a very strong constraint on RxDay occurrence across the WUS (Fig. S6). In most forested regions, consideration of 100-/1000-h FM results in 40–80 fewer RxDays per year; this difference highlights the importance of including larger-diameter FMs in prescribed fire planning. This effect is strongest in climatologically wet regions, namely: the Pacific Coast states of California, Oregon, and Washington, and locally also across the northern interior.

Whereas temperature, relative humidity, wind, and small-diameter FM are included in every burn plan reviewed for this study and consistently encountered by the authors in the field, 1000-h FMs are not consistently included in burn plans in the WUS. Yet our work shows that their inclusion has a large impact on potential RxDays in forested ecosystems. Critically, we find nearly ubiquitous projected decreases in 1000-h FM across all seasons and in nearly all regions, with the greatest decreases generally occurring in MAM and JJA (Fig. 5). (We note, however, that such decreases are only relevant in areas where heavy dead and down fuels are actually present; this generally excludes grassland and sagebrush-dominated ecosystems except in localized forested "fuel jackpots.") These widespread decreases in large-diameter dead FM and their contribution to undesirable fire effects (including difficult-to-control fire behavior and enhanced smoke production and carbon emissions[36]) highlight the importance of including metrics such as 100/1000-h FM and/or drought indices in burn prescriptions to better reflect long-term aridification trends amid a warming climate.

**Trends in air stagnation relevant to prescribed fire smoke emissions**. One additional meteorological consideration with respect to the practice and implementation of prescribed fire near highly populated areas is the potential contribution to local and regional air pollution. Although smoke emissions from prescribed fire are thought to be less harmful than from similarly sized uncontrolled wildfires[2], formal regulations set by regional air quality management entities as well as pressure from local communities regarding both real and perceived public health risks can result in substantial additional limitations[48] to implementing prescribed fire. Thus, we conduct an additional analysis examining simulated historical and future trends in lower atmospheric air stagnation (using near-surface winds and precipitation; henceforth, low-level stagnation, or "LLS").

We find widespread increases in projected LLS across the WUS in the 2021–2060 versus 1981–2020 period (Fig. 6). On a WUS-domain average basis across all seasons, LLS increases modestly (by +5%, or +5.7 days per year) between 1981 and 2060, with high ensemble agreement on the sign of change (16 of 18 model members depicting increases in LLS, Fig. S7). These increases are most widespread in DJF and SON (Fig. 6b, e) and most pronounced in the central and northern parts of the domain during JJA (Fig. 6d), and the Central Rocky Mountains and

adjacent Rocky Mountain Front Range during DJF (Fig. 6b). A notable regional and seasonal exception is a substantial JJA decrease in LLS in the central and western portion of the region affected by the North American Monsoon (i.e., southern Arizona and southern California; Fig. 6d). Thus, we find that meteorological changes to LLS due to climate change may bring about additional prescribed fire constraints in the current air quality regulatory environment.

**Discussion and conclusions**

Climate change has already resulted in more extreme wildfire burning conditions due to the warming and drying of vegetation across the WUS[23]. Despite these trends, there is widespread scientific agreement that a blanket fire exclusion policy will only exacerbate the problem[49] and a growing recognition surrounding the urgent need to decouple wildfire as a necessary ecosystem process from its sometimes catastrophic effects in a contemporary context. Increased fire on the landscape is inevitable in a warming climate, but there are active choices to be made regarding whether that fire comes in the form of increasingly high-intensity conflagrations—due to further increases in fuel loading, long-term vegetation aridification, and increasingly extreme fire weather conditions—or as the result of carefully planned, generally lower-intensity, and typically net-beneficial prescribed burning.

In this work, we find that climate warming is likely to bring about a substantial reduction in days meteorologically favorable for prescribed fire (i.e., RxDays) across the WUS—ranging from a ~1% decrease across the Northern Tier to a ~29% decrease in the Four Corners and a 24% decrease in the Pacific Southwest (Fig. 4) between 1981 and 2060. The strongest decreases in RxDays vary both regionally and seasonally but are generally weakest in winter (with some sub-regional increases) and most widespread in spring and summer (Fig. 3). Observed historical trends in RxDays between 1981 and 2020 are spatially heterogeneous, suggesting that more uniform decreases in RxDays may emerge imminently across much of the WUS as external climate forcing overcomes historical variability. Additionally, widespread projected decreases in large diameter FM across seasons suggest that vegetation aridity will become an increasingly important constraint on prescribed fire in a warming climate, especially in forested ecosystems (Fig. 5). We also find that the occurrence of smoke-trapping LLS events will broadly increase, potentially increasing air pollution-related concerns or further constraining burn windows (Fig. 6).

Collectively, these findings imply that, in addition to the many non-climatic barriers to prescribed fire implementation[31], climate change will likely further complicate efforts to use prescribed fire as a wildfire risk management and ecological enhancement tool. As has been evidenced by public policy conversations and real-world fire suppression decisions made in response to high-profile prescribed fire escapes[36,37], contemporary practitioners of prescribed fire are operating under climate and vegetation conditions that are increasingly outside the envelope of historical experience[32]. The narrowing of prescribed fire windows, as well as increases in extreme wildfire burning conditions at other times[47], will further challenge fire and land management agencies and entities already constrained by limited budgets and growing administrative burdens. This may be especially true across the Pacific Southwest (including California) and Four Corners regions—which are likely to see the largest declines in RxDays.

However, our findings also offer hope that prescribed fire can continue to be an invaluable risk management tool well into the future—and, with appropriate shifts in relevant policies and regulations, could potentially even be expanded in scope. Across much of the WUS, we find that winter either remains about as

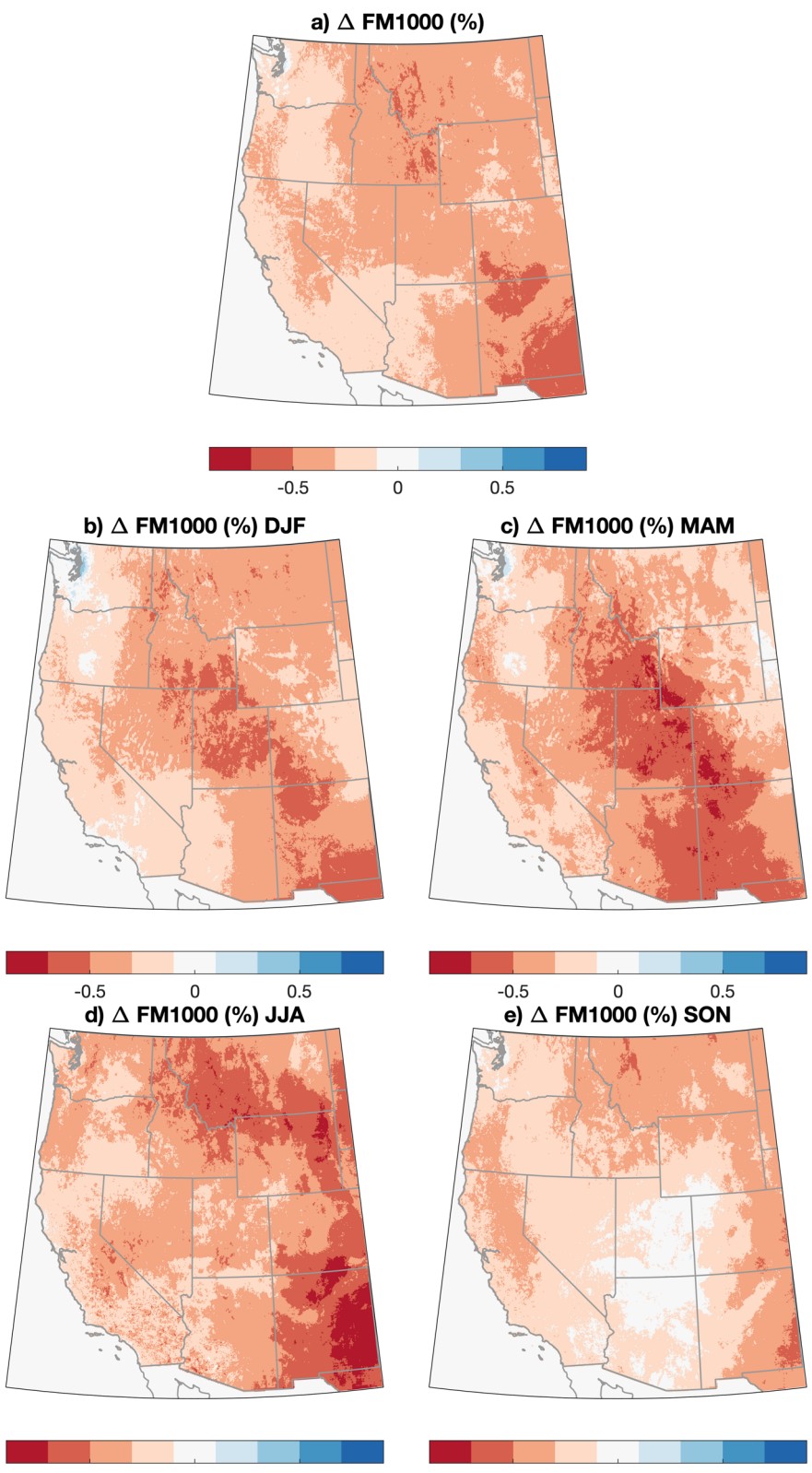

**Fig. 5 Maps of projected change in 1000-h dead fuel moisture across the WUS, 2021–2060 vs. 1981–2020.** Maps depicting the projected change in 1000-h dead fuel moisture across the western United States (WUS) on an annual (**a**) and seasonal (**b-e**) basis. The change is calculated as an 18-model average difference between 2021–2060 and the 1981–2020 period on a "moderate warming" (RCP4.5) trajectory using meteorological data from the downscaled CMIP5 climate model ensemble (MACA) dataset (1981–2060). Note: these values are only relevant in places where heavy dead and down fuels actually exist locally.

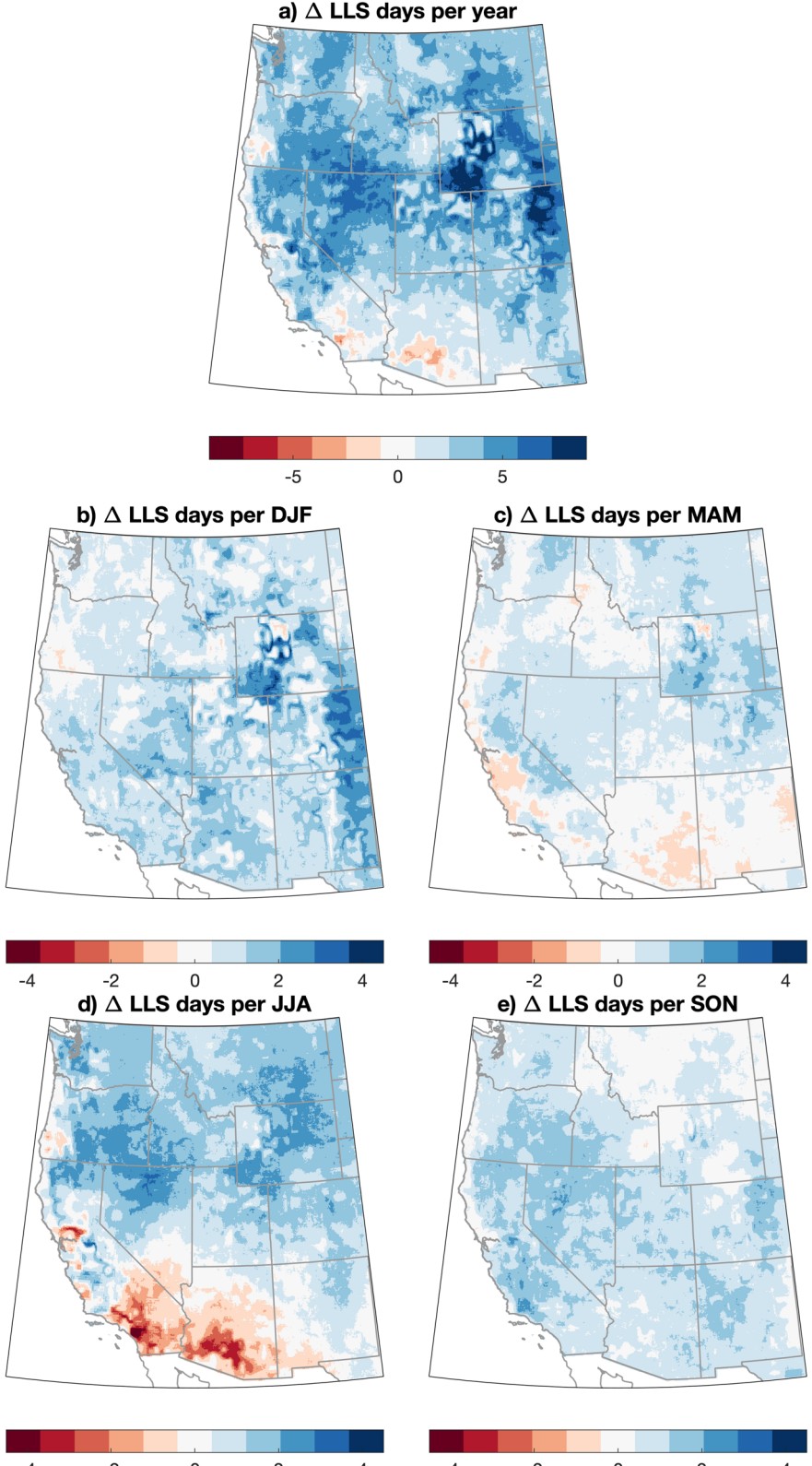

**Fig. 6 Maps of projected change in low-level air stagnation days across the WUS, 2021–2060 vs. 1981–2020.** Maps depicting the projected change in the number of low-level air stagnation (LLS) days across the western United States (WUS) on an annual (**a**) and seasonal (**b**–**e**) basis. Here, positive values (blue regions) represent areas expected to see an increase in low-level air stagnation days. The change in projected stagnation days is calculated as the difference between the air stagnation day counts for 2021–2060 vs. 1981–2020 period on a "moderate warming" (RCP4.5) trajectory using meteorological data from the downscaled CMIP5 climate model dataset (MACA).

favorable as during the late 20th century (Fig. 3) or even becomes more favorable across the Northern Tier. In some regions and/or seasons that were historically too moist and/or cool to support substantial prescribed fire, vegetation aridification may actually increase the overall number of RxDays. We also emphasize that even given projected declines, a majority of the WUS (especially non-forested locations) will continue to experience a substantial number of RxDays per year (Fig. 4).

Capitalizing on those remaining days still favorable for prescribed fire will require significant investment and policy changes to broaden presently narrow definitions of acceptable burn windows. In many parts of the WUS, for instance, prescribed fires are banned during spring out of concern for nesting birds and other protected species; elsewhere, seasonal fire personnel layoff dates are tied to the historical termination of autumn burn windows[32]. Additionally, seasonal wildland fire workers are often subject to limits on the cumulative time worked per year. Since seasonal employees currently make up the bulk of the workforce needed to implement burns, capitalizing on burn days outside of historical burn window timing—especially during winter, as our findings suggest will be increasingly important in a warming climate—would require a pronounced shift in agency fire crew staffing. Recent research focused on prescribed fire implementation in California ecosystems found that favorable meteorological conditions have indeed decreased in recent decades—but that winter and spring (DJF and MAM) have been historically underutilized due to staffing constraints that could potentially be alleviated via new administrative and policy choices[35].

Therefore, our findings provide direct evidence supporting recent calls for an expanded year-round fire management workforce whose responsibilities extend beyond fighting wildfires to also encompass the management of prescribed fire[50]. These findings also highlight the growing importance of tangible support—including increased funding and removal of existing regulatory barriers–for cultural burning practices by Indigenous fire practitioners, including via interagency partnerships[51]. Such burns have traditionally (for millennia) been tied to actual conditions and objectives rather than calendar dates[29]—and recent collaborations between tribal nations and various state, federal, and private entities have successfully returned winter burning in places where it had been largely absent in recent decades[28,52]. Our results also suggest a need to reevaluate smoke emission thresholds and constraints set by air quality regulators, as greater tolerance of dispersed smoke emissions from prescribed fires may potentially be deemed an acceptable trade-off to reduce the risk of extreme air pollution episodes resulting from catastrophic wildfire smoke events[6].

Ultimately, implementation of management, funding, and regulatory policies that enable greater flexibility by all kinds of "intentional burners" (i.e., prescribed fire and cultural burning practitioners collectively) would represent a climate adaptation commensurate with the projected shifts in WUS prescribed fire windows we find in this analysis. Since prescribed fires are likely to remain one of the most promising tools available to reduce potential wildfire hazards in a warming climate, we argue that reimagining prescribed fire implementation and policy to accommodate these evolving realities should be an urgent priority.

## Methods
### Defining and quantifying acceptable bounds for prescribed fire.
Prescribed fires are conducted across a wide range of ecosystems and across a fairly wide range of weather conditions. As such, the environmental envelope component of "burn prescriptions"—a predetermined suite of various acceptable vegetation and weather conditions conducive to safe and effective prescribed fire implementation––can sometimes vary considerably based on the local ecology, type, and density of vegetation present, the primary objectives of the burn (hazard reduction via fuels thinning, ecosystem restoration, etc.), and the entity performing the burn (the U.S. Forest Service, for example, may use considerably different sets of acceptable parameters for burning on public land than The Nature Conservancy does while burning on private land). Broadly speaking, burn prescriptions are intended to describe environmental conditions that yield vegetation combustion within an intensity range that is neither too low (due to damp vegetation, for instance, which would yield insufficient combustion intensity to achieve desired benefits) nor too high (extremely dry, hot, and/or windy conditions that might cause undesired high-intensity fire and potentially lead to an increased risk of fire spreading beyond intended boundaries).

In this analysis, we have assembled a range of real-world burn plans that were implemented by various federal, state, and private entities (including the US Forest Service, National Park Service, and The Nature Conservancy) between 2002 and 2022. These data were further separated into burn plans associated with predominantly forested landscapes and those associated with other landscapes (i.e., "non-forest" vegetation types). From this new dataset, we then calculate the median upper and lower prescription values for relevant weather variables (mid-flame wind speed, relative humidity, and air temperature) and composite fire danger indices (1-h FM, 10-h FM, and ignition component) across all available burn plans, and include 100-h and 1000-h FM from the subset of available plans in forest environments that included them.

The modest number ($n = 22$) of burn plans used in this study is justified due to the low variability in environmental parameters across all assimilated burn plans (even across regions and ecotypes). This stems from the fact that most or all contemporary prescription windows use the same underlying physical fire behavior models[53]. As biomass combusts and fires behave according to fundamental physical principles no matter the setting, the range of heat (in the form of temperature) and moisture (in the form of humidity and FM) requisite to achieve the desired level of combustion does not change across regions. Rather, our assessment of burn plans demonstrates that the timing and duration of such windows are the primary difference across biomes and regions—particularly when low-to-moderate intensity fire is the desired objective across most prescribed burns.

Separate thresholds were developed for forest and non-forest burn plans. Multiple environmental parameters exhibit modestly different ranges in forest versus non-forest plans (ST1), but the most notable difference is the absence of 100-h and 1000-h dead FM criteria in all non-forest plans (so such criteria are not included in our "non-forest" RxDay definitions). We apply these thresholds on a spatially explicit (grid box) basis, applying "forest" RxDay definitions on a spatially varying basis in locations classified as "forest" or "woodlands" by the Environmental Site Potential (ESP) product of LANDFIRE[54] as in previous work[9] and applying "non-forest" RxDay definitions in all other locations.

Finally, we create a spatially explicit dataset using corresponding daily average data from gridMET, and the CMIP5 RCP4.5 downscaled ensemble simulations to determine whether a particular gridbox is "in prescription" or not in a given calendar day, depending on whether it falls within the upper and lower median values for all of the above-identified metrics. We use this binary dataset of locations "in prescription" or "not in prescription" in the remainder of the analysis.

We recognize that the creation of a composite prescribed fire metric from different burn plans in different ecosystems is a compromise; as noted above, real-world burn plans must account for a high degree of sub-regional complexity in

vegetation, background climate, and aridity, guidance and mandates by supervising entities, and other considerations. As such, our use of a single set of values across the entire WUS represents a substantial (though necessary) simplification. We also emphasize that prescribed fire is not appropriate in all ecosystems—particularly arid deserts, where large-scale fires were historically rare, and many plant species are not fire-adapted. However, we re-emphasize that the primary goal of all prescribed burn plans is to allow for vegetation combustion within a certain range, attempting to prevent either very low or very high intensity–conditions that are broadly bookended by the range of weather and vegetation aridity parameters included in our composite metric (described further in the following paragraph).

In this analysis, the quantitative framework we use to define a composite prescribed fire metric (i.e., to determine whether conditions, in aggregate, are within acceptable bounds for prescribed fire in a given location on a given day) is based upon a suite of near-surface meteorological variables (including air temperature, relative humidity, and wind speed) in addition to fire metrics from the US National Fire Danger Rating System (NFDRS[55]) (ignition component as well as 1-h, 10-h, 100-h, and 1000-h dead FM). Although NFDRS is generally used by fire managers to track broader trends in seasonal wildfire risk, we use it to derive these specific variables because they are also often included in burn prescriptions. A comprehensive list of all relevant data thresholds used in these calculations is provided in ST1). When all variables are within an acceptable range (that is, bounded by values that are neither too low nor too high), we determine that a given location is "in prescription" and that a "prescribed fire day" (RxDay) has occurred. Therefore, the definition used here is a binary one. We subsequently assess the background seasonality as well as historical and projected future trends on the basis of regional and seasonal counts of RxDays.

An important caveat regarding our composite RxDay definitions is that this bulk metric cannot be, and is not intended to be, fully representative of highly localized microclimate variations—nor does it capture various ecological, legal, and/or regulatory constraints that can and do dictate on-the-ground decisions regarding whether to proceed with a particular treatment on a given day. As such, the RxDay values and trends discussed in this analysis cannot be directly compared to real-world implementation of prescribed fire in specific locations. Any real-world treatment would require detailed local weather and vegetation information that is simply not available even at the horizontal resolution (i.e., ~16 km$^2$) of the downscaled climate model data used in this study. Instead, our goal here is to assess broad-scale trends in meteorological conditions relevant to prescribed fire across the WUS while acknowledging that local definitions and realities may differ considerably and in consequential ways on specific prescribed burns.

**Additional information regarding specific burn plans used in this analysis**. We compile a total of 22 burn plans from across the WUS domain for use in this analysis. Fourteen of these plans are from primarily forested landscapes, and 8 are from primarily non-forested landscapes. The geographic distribution of the plans extends widely across the study domain, including 12 plans from California, 2 each from Colorado, Idaho, and Nevada, and 1 each from Washington, Oregon, Wyoming, and New Mexico (Fig. S1). The ranges of environmental conditions contained in forest versus non-forest plans are included in Table S1, and the specific parameters used in each plan may be found in the public repository noted in the "Data availability" section[56].

**Testing the sensitivity of baseline and trends to underlying RxDay definitions**. As noted above, we use large-scale vegetation type-aware definitions for RxDays (i.e., forest versus non-forest regimes) that vary across the landscape according to the predominant vegetation type in a particular 4 × 4 km grid box. To systematically test the sensitivity of both RxDay baselines and historical trends, we compare three possible RxDay definitions: first, one in which all available burn plans (covering forest and non-forested regions) are used to define acceptable ranges for environmental parameters; second, one in which only those burn plans from forested landscapes are used; and third, one in which only those burn plans from non-forested landscapes are used. We then apply these definitions uniformly across the landscape, irrespective of the actual vegetation type in a particular location, to explore the widest plausible parameter space from among our sample of real-world burn plans. Specific differences between the environmental conditions using each of these definitions can be found in ST1, though the most notable difference is that FM100/1000 are not included in non-forest plans.

We find that there is high sensitivity to the use of forest versus non-forest plans with respect to annual RxDay baseline occurrence (around 66 RxDays per year on a domain-wide average using non-forest definitions uniformly across the landscape, and 24 RxDays per year applying forest definitions uniformly across the landscape; Fig S8), which is expected given the complete exclusion of large-diameter FMs from non-forest plans. However, we find low sensitivity to the use of different RxDay definitions to historical trends in annual RxDays on a domain-wide basis (a loss of 7 RxDays per year using a forest definition and a loss of 7.2 RxDays per year using a non-forest definition; also, note the spatially similar WUS trends in Fig. S9, with strong RxDay declines in the Pacific Southwest and Four Corners regions but little change or even slight increases across the Northern Tier). This indicates that key study findings are not strongly sensitive to RxDay definitions across the plausible range of values derived from real-world burn plans.

**Data sources and climate change scenarios considered**. For observed historical trends (1981–2020), we use the 4-km grid-MET dataset[57]. For simulated historical (1981–2020) and future (2021–2060) projections, we utilize climate model data statistically downscaled to a 4-km spatial resolution using the Multivariate Adaptive Constructed Analogs (MACA) method with gridMET data serving as the training dataset for downscaling from the original GCM resolutions ("MACAv2-METDATA"; https://www.climatologylab.org/maca.html). The first ensemble member of 18 individual climate models (ST2) from the Coupled Model Intercomparison Project Phase 5 (CMIP5) is used to compute multi-model averages for a suite of physical variables (ST1). For the 1981-2005 period, we use the "Historical" CMIP5 climate forcings (which include both anthropogenic greenhouse gas emissions and anthropogenic aerosol emissions); for the 2006-2060 period, we use a "moderate emissions" climate change scenario (RCP4.5), which represents a greenhouse gas emissions trajectory that would most likely produce around ~2 C of global mean warming by ~2060[58], which is lower than warming likely to result from current international climate policy.

Rather than use daily average conditions for directly assessing whether meteorological variables were in prescription, we evaluate temperature, relative humidity, and wind speed representative of peak burning conditions that would most likely materialize during the afternoon hours on potential RxDays. Specifically, we use daily maximum temperature and daily minimum relative humidity, and we scale wind speeds by a factor of 1.5 (so that they are 50% higher than daily average

values to account for the greater near-surface wind speeds during the afternoon). Finally, since prescribed fire burn plans require wind speeds at "mid-flame" height, we scale the 10 m wind speed by a factor of 0.4 to account for surface frictional effects[59].

To calculate NFDRS metrics (ignition component, 1-h, 10-h, 100-h, and 1000-h dead FM), we primarily use traditional formulas originally described by[55]. For all NFDRS metrics calculated using downscaled climate model ensemble (CMIP5/MACA) data, we apply a secondary bias correction to account for biases arising from covariance and serial correlation in the underlying data (e.g.,[60]) using a quantile mapping approach (i.e., comparing downscaled MACA data to the original gridded metrics from gridMET pseudo-observations).

**Projected air stagnation trends**. We again use statistically downscaled climate model data (MACA, as described above, with Historical + RCP4.5 forcings) to calculate a lower-atmospheric air stagnation metric using daily averaged near-surface (10 m) wind speed of <3.2 m/s and daily accumulated precipitation of <1 mm for the entire 1981–2060 period. These variables and associated thresholds comprise 2 of the 3 criteria that make up the air stagnation index (ASI), and we use a modified version of this index focused on daily quantities following[61]. Although this metric does not account for local topographic effects on smoke dispersion, it broadly captures conditions that would inhibit ventilation at regional scales. We focus on the near-surface aspects of potential air stagnation (the third variable in the ASI is mid-tropospheric [500 mb] winds) as we deem the strength of low-level ventilation (via surface winds) and rate of scrubbing of particulate matter from the atmosphere (via scavenging by falling hydrometeors) to be most relevant in determining the potential public health impacts associated with smoke generated by pre-scribed burning. We also note that air quality indicators are used in widely varying ways in different regions and under different administrative jurisdictions to manage particulate matter pollution risk, so our use of the LLS in this context is an indirect proxy for air quality constraints rather than an explicit representation of regulatory constraints on prescribed fire.

**Data analysis and visualization**. All time series are generated using aerial averages for the region encompassing (latitude range 31.3°N–48.9°N; longitude range 235.3°E–256.9°E) using all non-ocean grid points within the boundary of the continental United States. We also define three sub-regions ("Pacific Southwest," approximately encompassing California and Nevada (32°N–42°N, 235.3°E–246°E); the "Northern Tier," approximately encompassing Oregon, Washington, and Idaho, as well as most of Montana and Wyoming from the Rocky Mountain Front Range westward (42°N–48.9°N, 235.3°E–256.9°E); and the "Four Corners," approximately encompassing Arizona and Utah, as well as most of Colorado and New Mexico from the Rocky Mountain Front Range westward (31.3°N–42.0°N, 246.0°E–256.0°E). The time series depicted in Fig. 4 represents 5-year moving averages. We also calculate a simple linear trend for the WUS-prescribed fire days using linear regression and assess the statistical significance (p-value) of these regionally and seasonally specific regressions using a two-tailed test.

**Caveats surrounding climate change scenario and vegetation feedbacks**. We emphasize that global warming greater than depicted in the RCP4.5 scenario (which results in around +2 C of global warming by 2060, below "current policy" warming estimates of +2.3 to +3.0 C by 2100[58]) or accelerated adverse ecosystem-level vegetation responses to warming (via, for example, large scale forest mortality and/or type conversion[19]) would

likely yield actual decreases in prescribed fire opportunities greater than quantified in this analysis.

## Data availability

Existing datasets used in this analysis may be accessed via the web. GridMET historical observations of weather and vegetation conditions are available at: https://www.climatologylab.org/gridmet.html, and downscaled CMIP5 data (MACA) are available at: https://www.climatologylab.org/maca.html. New data created as part of this analysis, including gridded RxDay counts and documents describing all real-world burn plans used in defining the RxDay composite metric, are publicly archived on Zenodo and can be accessed at https://doi.org/10.5281/zenodo.7864935.

## Code availability

Code used in the analysis may be obtained upon request via the corresponding author.

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

## Acknowledgements

We thank the Editor, as well as three anonymous reviewers, for their thoughtful feedback, which has substantially improved the manuscript. D.L.S. received support through a collaboration between the Institute of the Environment and Sustainability at the University of California, Los Angeles; the Center for Climate and Weather Extremes at the National Center for Atmospheric Research; the Nature Conservancy of California; and NSF award 1854761. J.T.A. received support from NSF award OAI-2019762. D.A.K. was supported by NASA FINESST award 80NSSC21K1603.

## Author contributions

Conceptualization: D.L.S., J.T.A., C.K., K.S., D.A.K., D.S., and E.S. Methodology: D.L.S., J.T.A., C.K., K.S., D.A.K., D.S., and E.S. Data acquisition and curation: D.L.S., J.T.A., C.K., K.S., D.A.K., D.S., and E.S. Investigation: D.L.S., J.T.A., and D.A.K. Visualization: D.L.S., J.T.A., and D.A.K. Writing—original draft: D.L.S. Writing—review and editing: D.L.S., J.T.A., C.K., K.S., D.A.K., D.S., and E.S. Project administration: D.L.S.

## Competing interests
The authors declare no competing interests.
