## [Peer Review File · Communications Earth & Environment]

3rd Apr 23

Dear Dr Swain,

Your manuscript titled "Climate change is narrowing and shifting prescribed fire windows in western United States" has now been seen by 2 reviewers, whose comments are appended below. You will see that they find your work of some potential interest. However, they have raised quite substantial concerns that must be addressed. In light of these comments, we cannot accept the manuscript for publication, but would be interested in considering a revised version that fully addresses these serious concerns.

We hope you will find the reviewers' comments useful as you decide how to proceed. Should additional work allow you to address these criticisms and meet the editorial thresholds outlined below, we would be happy to look at a substantially revised manuscript. In the following we list our editorial thresholds, based on reviewer comments:

- Provide compelling new insights into how climate change influences observed historical and projected trends in the number and timing of suitable days for prescribed burning in the Western USA;
- Provide comprehensive method details and clearly discuss the limitations of your analysis and the underlying assumptions;
- Include uncertainty analyses that indicate the sensitivity of your results to certain assumptions and demonstrate that your findings are robust;
- Consider including a comparison of historical available burn days with actual burn days.

If you choose to take up this option, please either highlight all changes in the manuscript text file, or provide a list of the changes to the manuscript with your responses to the reviewers.

If the revision process takes significantly longer than three months, we will be happy to reconsider your paper at a later date, as long as nothing similar has been accepted for publication at Communications Earth & Environment or published elsewhere in the meantime.

We understand that due to the current global situation, the time required for revision may be longer than usual. We would appreciate it if you could keep us informed about an estimated timescale for resubmission, to facilitate our planning. Of course, if you are unable to estimate, we are happy to accommodate necessary extensions nevertheless.

Please use the following link to submit your revised manuscript, point-by-point response to the reviewers' comments with a list of your changes to the manuscript text (which should be in a separate document to any cover letter) and any completed checklist:

[link redacted]

Please do not hesitate to contact me if you have any questions or would like to discuss the required revisions further. Thank you for the opportunity to review your work.

Best regards,

Clare Davis, PhD
Senior Editor
Communications Earth & Environment

www.nature.com/commsenv/
@CommsEarth

EDITORIAL POLICIES AND FORMAT

If you decide to resubmit your paper, please ensure that your manuscript complies with our editorial policies and complete and upload the checklist below as a Related Manuscript file type with the revised article:

Editorial Policy Policy requirements (Download the link to your computer as a PDF.)

For your information, you can find some guidance regarding format requirements summarized on the following checklist:(<https://www.nature.com/documents/commsj-phys-style-formatting-checklist-article.pdf>) and formatting guide (<https://www.nature.com/documents/commsj-phys-style-formatting-guide-accept.pdf>).

REVIEWER COMMENTS:

Reviewer #1 (Remarks to the Author):

This manuscript illustrates how common environmental conditions used for planning prescribed burns in the western US are distributed over the past 30 years and how they are projected to change with a 2°C rise in global temperature (RCP4.5). This manuscript joins other recently published works assessing how prescribed burning—a tool critical for wildfire risk reduction and resilience of fire adapted communities—is likely to be affected by projected climate change. With major revisions, this manuscript should be acceptable for publication.

1) Generally, this reviewer finds the logic of the introduction to be flawed and suggests it be revised more in line with the discussion. As written, it undercuts the regional relevance of the work. This

concern is best captured in Para. 2 with an immediate climate change focus being out of place relative to the consensus that there is a problem of insufficient “beneficial fire” in the WUS. Does climate change currently constrain prescribed fire activity? A quick glance at NIFC data suggests no (it has not declined), but climate-as-major impediment to prescribed fire in the “present” is implied throughout the intro which undermines the relevance of the finding that future climate may further limit prescribed fire activities. Put another way, the introduction appears to present climate as the dominant challenge of prescribed fire in the past 20 years, and as such, a focus on continued loss of prescribed fire opportunity is moot, since our current pace and scale is clearly not impacting hazard now. This reviewer arrives at this conundrum around lines 67-68. Why is this work relevant in western regions that don’t utilize enough prescribed fire to matter now? With the growing consensus that the West needs more beneficial fire (thus why one would pursue analysis of prescribed burn windows), then presenting climate as a dominant barrier currently to increasing pace and scale is self-defeating, then why try? With a future that is worse, prescribed fire is simply a boutique tool with little adaptive capacity? The discussion does a much better job of presenting a logical justification for the study’s relevance.

2) By presenting the dominant problem in the west being simply too little beneficial fire, which had led to too many trees and too much fuel, then burn windows as a climate challenge are properly in context. That problem of fuel accumulation (particularly large diameter wood) and fire adapted communities can then intersect with too many people in an aridifying landscape as the authors have well documented in their past work. It would also allow the authors to justify the 1000-hr analysis and cite what prior work done on coarse woody debris and risk has been done (this work is missing when the topic of 1000-hr fuel is first brought up in the last para. of the Introduction).

3) Fundamental to fire in the Western US is the lack of prescribed burning on currently available burn days. Reliance on broad burn plans to define windows but not assessing with current utilization of those windows obviates the trends in burn day availability. How poorly managers utilize those days currently should inform future modeling interpretation of impacts in this study. The maps of figure 1 are interesting in this regard as NM and AZ show among the fewest burn days but consistently have among the highest prescribed fire acres in the west. While outside the current scope of the modeling in this study, this comparison with actual burn days is HIGHLY relevant to interpretation. As presented, how can the authors assess impact without data on when burns are conducted? This is made more important by the broad window applied to such a large geographic area (WUS)—are some states/regions utilizing all their current burn windows?

4) The results appear to present some methods creating confusion and duplication.

5) It appears that methods followed similar methods to recent studies (Baijnath-Rodino et al. 2022, Kupfer et al. 2020), but those didn’t attempt to apply the composite binary “burn window” across such a broad geography. The burn plan is not a prescription (the environmental window which is an element in the standard burn plan used by the Agencies and TNC references in methods), and is mistakenly confused with the “prescription” in this study. Again, a simple attempt to relate the burn window to actual burn activities is suggested. Those records should be available from agency prescribed fire reports for the historical time of interest.

6) Lastly, critical details must be added to the methods on the burn plans used: How many burn plans were used to develop the prescription window and produce a map of their geographic coverage in S1. This reviewer could not find this information and suggests that actual burn plans

used also be archived with supplemental information (currently unavailable to this reviewer to assess the geographic distribution or relevance of burn prescription details across such a broad region). Given the broad regional differences in burn plan description of burn windows and objectives, the lack of subregional criteria is problematic to the overall analysis.

Minor Comments:

Abstract

Line 26: “Purposeful burning” is not an appropriate definition of prescribed fire in this context, and the description that it is “challenging” to implement is only regional. Arson is purposeful.

The abstract while beginning with a focus on the west loses that regional context quickly, which should be clarified in summary of results since these findings mirror other recent studies on the topic.

Introduction

Line 36: With wildfire crisis in quotes, there should be a citation, presumably the USFS report of that title.

Line 54: use a standard definition of prescribed fire. Restoring precolonization processes is not an ecologically sound objective for prescribed burning nor language commonly used for current burn plans or ecological objectives. Increasingly, resilience is the objective in light of invasive species and climate change. restoring to past processes is inconsistent with the modeling results.

Line 69: It seems odd to present summary results here, and there is repetition from the Abstract.

Line 74-75: This statement on 1000-hr fuels is incorrect and must be removed. If a focus of the paper, it must be given attention in the introduction as there is considerable literature on the topic. The importance of this fuels in escape risks has been recognized for decades in fire management both in wildfire and prescribed fire containment. Considerations of smoldering fuels is implied in burn plan reliance on KBDI and other Drought indices that also incorporate other smoldering fuel potential, including organic soil horizons (e.g., duff). See problems with analysis of 1000-hr fuels regionally in Figure comments.

Results

Line 216. One of the reports cited here relevant to coarse woody debris is from the 2000 Cerro Grande escape, further emphasizing the long-term realization of its importance in prescribed fire risk analysis (see above). There are additional studies of downed and dead fuels from across the west and their growing contribution to fire behavior. Furthermore, this study relies on modeled values here of putative drying dynamics of 1000-hr fuels with respect to their moisture dynamics. Those empirical data are from sound wood may not be relevant in an aridifying west with an abundance of decaying fuels from regional bark beetle die off a decade ago.

Figure 5. The inclusion of extensive sagebrush and short grass prairie in the 1000-hr dead fuel moisture calculation is inappropriate. Those systems (e.g., eastern MT, large portions of WY, and eastern NM, do not support forested vegetation relevant to this graphic, which is very misleading as those areas are shown to have the greatest potential change in 1000-hr fuel moisture.

Discussion

Line 272: The use of “purposeful” should be replaced by prescribed fire (see comment above). If

trying to include the expansion of managed wildfires for objectives, be consistent with policy terms in the region rather than the vague term “purposeful.”

Methods

It must be mentioned that time lag fuels used in prescribed fire planning and NFDRS are constructs used primarily for wildfire behavior under extreme conditions. The metrics while common have little to no broad association with prescribed fire planning and thus drought indices are more commonly used as a surrogate (but only included in prescriptions through burn planner experience or regional rules of thumb). In the eastern US and portions of CA, for example, the range of acceptable 1000-hr fuel moistures in S-1 (10-20%) would represent unacceptably significant risk of smoldering consumption.

References

Is “Fire, C.” the proper citation?

Reviewer #2 (Remarks to the Author):

Review of the article: Climate change is narrowing and shifting prescribed fire windows in western United States, by Swain et al.

General comments:

The manuscript is well written and addresses a very important problem - i.e., how Rx burn windows might change in the western US. I have some general broad-level questions and concerns. Therefore I recommend major revisions.

How different would the analysis be if you used local prescriptions for what constitutes one Rxday? I would recommend conducting an uncertainty analysis if you used spatially different burn prescriptions.

I am not fully sure about the LLS analysis. It is of course dependent on other factors such as wind, terrain, and local microclimatic situations. How well is it captured and how is it explicitly considered in the burning window prescription?

How reliable are the estimates for 1,10,100 and 1000 hr fuel moisture?

I think more information is needed to understand how to estimate the optimum Rx window. Can you provide the different Rx burn prescriptions used in the study? Which prescriptions are used in which region? How do you know that the prescription you derive is optimal?

Swain et al. 2023 Response to Reviewers

We thank both reviewers for their feedback and constructive criticisms. We have conducted new analyses and modified the manuscript substantially in response to these comments, which are enumerated in detail below.

This Response to Reviewers document provides a complete documentation of all changes made in response to the reviews. The document is designed so that the changes that we have made in response to each comment can be immediately read and understood, independent of the other comments and responses.

Reviewer comments are shown in **bold**. Author responses are shown in plain text. Passages excerpted from the main manuscript are *italicized*. In the revised manuscript, all changes made in revision are noted using **red** text to aid in quick reference.

Reviewer #1 Remarks:

This manuscript illustrates how common environmental conditions used for planning prescribed burns in the western US are distributed over the past 30 years and how they are projected to change with a 2°C rise in global temperature (RCP4.5). This manuscript joins other recently published works assessing how prescribed burning—a tool critical for wildfire risk reduction and resilience of fire adapted communities—is likely to be affected by projected climate change. With major revisions, this manuscript should be acceptable for publication.

We thank the reviewer for their comprehensive and thorough assessment, and are glad the reviewer believes our manuscript should be acceptable for publication following revisions. We have made major changes to the text as well as conducted substantial new analysis to address the feedback raised by the reviewer, which we feel have greatly strengthened the manuscript.

1) Generally, this reviewer finds the logic of the introduction to be flawed and suggests it be revised more in line with the discussion. As written, it undercuts the regional relevance of the work. This concern is best captured in Para. 2 with an immediate climate change focus being out of place relative to the consensus that there is a problem of insufficient “beneficial fire” in the WUS. Does climate change currently constrain prescribed fire activity? A quick glance at NIFC data suggests no (is has not declined), but climate-as-major impediment to prescribed fire in the “present” is implied throughout the intro which undermines the relevance of the finding that future climate may further limit prescribed fire activities. Put another way, the introduction appears to present climate as the dominate challenge of prescribed fire in the past 20 years, and as such, a focus on continued loss of prescribed fire opportunity is moot, since our current pace and scale is clearly not

impacting hazard now. This reviewer arrives at this conundrum around lines 67-68. Why is this work relevant in western regions that don't utilize enough prescribed fire to matter now? With the growing consensus that the West needs more beneficial fire (thus why one would pursue analysis of prescribed burn windows), then presenting climate as a dominant barrier currently to increasing pace and scale is self-defeating, then why try? With a future that is worse, prescribed fire is simple a boutique tool with little adaptive capacity? The discussion does a much better job of presenting a logical justification for the study's relevance.

We thank the reviewer for this thoughtful response, and are glad that the reviewer believes we have framed the problem well in the discussion. We agree with the reviewer that the tone of the introduction is somewhat different than that in the discussion with respect to contextualizing the relative role of climate versus other non-climate factors, and also agree that we may have initially overemphasized the degree to which climate change has constrained prescribed burning historically.

In response to this feedback, we have substantially re-written large sections of the introduction to clarify most of the points the reviewer raises in the comment, and to generally align the introduction better with the discussion and conclusions section. We do note that, due to space constraints in the introduction and the fact that the discussion section is generally viewed as a more appropriate place to offer evidence-guided exposition and informed opinions than the introduction section, the tone of these two sections remains somewhat different. Despite this, we feel that the revised introduction now more accurately contextualizes the historical role of climate and prescribed fire, more clearly highlights the fact that existing burn windows are generally underutilized, and even more strongly emphasizes (as the reviewer correctly points out) the growing consensus regarding the widespread need for beneficial fire in the Western U.S.

We do note that looking for declining prescribed fire usage in NIFC data is an inadequate test for climate change as a limiting factor, given that NIFC has sought to but struggled to increase prescribed fire in the WUS for a broad range of reasons over the last four decades. It is well documented that climate change, specifically through extreme and sometimes record-breaking fire weather events yielding escaped prescribed fires, has set burgeoning prescribed programs back and produced national prescribed fire moratorium periods (Kolden and Brown 2010), and that changing burn windows have been cited by prescribed fire personnel in interviews as a challenge to increasing its use, despite national pressure to do so (Kolden and Brown 2010, Schultz et al. 2019). There is also substantial evidence that climate change is lengthening fire seasons in parts of the western US, which in turn limits firefighter availability in areas that are in prescription for lighting "good fire", an example of which resulted in the Caples escaped prescribed burn in 2019 (<https://tinyurl.com/mpe5vuzh>). Moreover, by the time fire season dies down across broader areas, most seasonal federal firefighters, who are needed to staff the burns,

are reaching their maximum number of seasonal hours, which limits how much prescribed fire agencies can accomplish. We have added text to the Introduction to emphasize that there may be additional indirect effects from climate change (i.e., beyond those directly affecting meteorological parameters on potential burn days) even where/when climate has not played a dominant role at first glance.

We now enumerate specific changes to the text in response to the feedback above.

To contextualize historical climate change vs. non-climate factors relevant to wildfire (please note that the entire third paragraph of the manuscript/introduction is devoted to highlighting that climate change is far from the only factor at play with respect to historical trends, and that non-climate factors can be of comparable or even greater relevance locally):

“Recent work points toward anthropogenic climate change rapidly altering Western United States (WUS) fire regimes to produce overall more extreme wildfires⁸⁻¹² that burn over a longer fire season^{13,14}, at higher elevations¹⁵, and with greater synchronicity across multiple regions^{16,17}, ultimately producing more smoke and greater carbon emissions¹⁸. Although most ecosystems across the western US are fire adapted, recent increases in the severity and frequency of wildfire have become disruptive—causing the loss of old-growth forest¹⁹, ecosystem type conversions²⁰, and reductions in carbon storage²¹.

However, even as projections suggest future warming will further amplify these trends (e.g., ²² and ¹⁷), other critical non-climate factors are driving profound changes in wildfire regimes, each with different near-term local solutions²³. These stressors include excessive fuel accumulation in some ecosystems due to 20th century fire exclusion^{24,25}, changing patterns of human-caused ignitions²⁶, and the expansion of populated areas into high-risk zones²⁷.” (Lines 48-59)

To emphasize the dominant importance of non-climate factors in prescribed fire historically, and that existing burn windows are widely underutilized as well as to better motivate why we have chosen our particular research question in this context:

“Although considered a widely applicable solution, there are many impediments to prescribed fire implementation^{30,31}, including staffing and funding limitations, risk tolerance and smoke impacts³²). For these and other reasons, prescribed fires are not implemented during all suitable burn windows—suggesting that, to date, climate has not been the primary inhibitor to implementation and that present-day burn windows are often underutilized³³⁻³⁵. However, in recent years, the combined effects of severe short-term drought and long-term aridification¹⁶ have contributed to a reduction of adequate spring and autumn burn windows in some regions^{32,35}, raising concerns that climate change will add to the many existing challenges to prescribed fire implementation³⁶.” (Lines 84-92)

And:

“Given calls for a substantial expansion of prescribed fire implementation to combat wildfire risk^{30,39,40}, there is a critical need to understand the extent to which climate change might alter the seasonality and frequency of burn windows. To significantly increase prescribed fire implementation, the other impediments to prescribed fire will need to be addressed; however, understanding potential shifts in burn windows could empower state and federal agencies to create more realistic staffing plans that maximize the potential for prescribed fire implementation.” (Lines 104-110)

Finally, to emphasize indirect reasons in which climate change may have affected historical implementation of prescribed fire:

“Beyond direct meteorological constraints, climate change has already demonstrated indirect impacts on prescribed fire implementation. Extreme meteorological events, especially severe to historically unprecedented drought condition, have been contributing factors to several prescribed fires that “escaped” and became disastrous wildfires. The societal and political fallout from such events has led to multiple temporary U.S.-wide moratoria on prescribed fire (such as after the 2000 Cerro Grande Fire in New Mexico³⁷ and the Hermits Peak Fire in 2022³⁸. Longer fire seasons also mean that fire personnel are committed to fighting fire in some regions—which in turn reduces the number of trained personnel available to implement burns in other regions where conditions are favorable. In addition, longer fire seasons are particularly challenging for federal agencies since their firefighting workforce is dominated by seasonal employees who can only work for limited durations³³⁻³⁵.” (Lines 93-103)

We have also added a sentence clause in the discussion to explicitly drive this point home when discussing potential solutions:

“Collectively, these findings imply that, in addition to the many non-climatic barriers to prescribed fire implementation³¹, climate change will likely further complicate efforts to use prescribed fire as a wildfire risk management and ecological enhancement tool.” (Lines 350-352)

2) By presenting the dominant problem in the west being simply too little beneficial fire, which had led to too many trees and too much fuel, then burn windows as a climate challenge are properly in context. That problem of fuel accumulation (particularly large diameter wood) and fire adapted communities can then intersect with too many people in an aridifying landscape as the authors have well documented in their past work. It would also allow the authors to justify the 1000-hr analysis and cite what prior work done on

coarse woody debris and risk has been done (this work is missing when the topic of 1000-hr fuel is first brought up in the last para. of the Introduction).

We thank the reviewer for this feedback. We have added language to reflect existing work related to 1,000 hour fuels to the Introduction and methods, and better highlighted the relevance of 1000-hr fuels to prescribed burn windows in the precise location in the Introduction the reviewer identifies:

“We also explore and highlight the potential importance of accounting for large-diameter fuel moisture as a proxy for the impact of long-term drying trends on woody fuels (i.e., primarily in forested ecosystems), which to date has only been included in a limited number of WUS burn plans. These large diameter (i.e., 1,000-hr) fuels have also become increasingly important in a fuels management context following mass tree die-off events due to drought and bark beetles in this region^{41,42}, and in the context of managing smoke-related air quality concerns.” (Lines 117-123)

Additionally, in response to the above comments, we have removed use of the term “coarse woody debris” (CWD) and now explicitly refer to different diameter fuel moistures were relevant.

We would also like to clarify that none of the authors has in fact suggested either here or in prior work that there are too many people in an aridifying WUS; rather, we have simply conducted research documenting the impacts of increasing population on fire patterns in this region.

Finally, as discussed in the response to other comments from the reviewer, we have extensively re-written other sections of the Introduction to better align with the Discussion (in which the role of long-term aridification is directly contextualized relative to other relevant factors).

3) Fundamental to fire in the Western US is the lack of prescribed burning on currently available burn days. Reliance on broad burn plans to define windows but not assessing with current utilization of those windows obviates the trends in burn day availability. How poorly managers utilize those days currently should inform future modeling interpretation of impacts in this study. The maps of figure 1 are interesting in this regard as NM and AZ show among the fewest burn days but consistently have among the highest prescribed fire acres in the west. While outside the current scope of the modeling in this study, this comparison with actual burn days is HIGHLY relevant to interpretation. As presented, how can the authors assess impact without data on when burns are conducted? This is made more important by the broad window applied to such a large geographic area (WUS)—are some states/regions utilizing all their current burn windows?

We strongly agree with the reviewer that systematically understanding how many burn days are actually utilized across the entire Western United States is an important line of inquiry, and would make for a very interesting analysis. However, as the reviewer notes, that would be well beyond the scope of this study as comprehensive data on dates, locations, and objectives of realized prescribed burns are not readily available. It has been well documented that record keeping of prescribed burning is poor at best, and records kept are not readily available nor appropriate for scientific research. PFIRS is meant to be the most comprehensive database of prescribed fire, but it only catalogs prescribed burns in California. Burns across the rest of the WUS are not reported systematically with daily and geospatial accuracy, which is what would be required for such an analysis.

We recognize that it is possible that such data is less accessible for WUS prescribed fires than elsewhere in the United States, but after considerable research it does not appear that such data is actually available systematically, and it would require a large (and possibly in-person) effort to obtain records (which are sometimes not digitized) maintained by a wide range of different entities.

As noted elsewhere, we also agree with the reviewer that many potentially favorable burn days go unutilized even in the present climate for various non-meteorological reasons. To emphasize this more strongly and to ensure our readers from outside prescribed fire circles are fully aware of this reality, we have added the following text to the manuscript that states so directly:

“Although considered a widely applicable solution, there are many impediments to prescribed fire implementation^{30,31}, including staffing and funding limitations, risk tolerance and smoke impacts³²). For these and other reasons, prescribed fires are not implemented during all suitable burn windows—suggesting that, to date, climate has not been the primary inhibitor to implementation and that present-day burn windows are often underutilized³³⁻³⁵.” (Lines 84-88)

4) The results appear to present some methods creating confusion and duplication.

We appreciate this feedback. After reviewing the results subsections, we recognize that there are indeed multiple occasions on which brief explanations of the metrics being discussed (and how they are calculated) are given, followed by a parenthetical note to “See Methods/See Methods for additional details”. Given the structure of our analysis and the importance of specific definitions to the interpretation of results, we are not sure that removing these brief (generally 1 sentence or shorter) explanations from the Results section would be possible without making it much harder for the typical reader to interpret our paper. Although this does mean there is a small amount of duplication between this section and the Methods, we feel this is preferable to leaving the reader

in the dark regarding where our data originates or the broad assumptions we have made in calculating trends.

5) It appears that methods followed similar methods to recent studies (Bajjnath-Rodino et al. 2022, Kupfer et al. 2020), but those didn't attempt to apply the composite binary "burn window" across such a broad geography. The burn plan is not a prescription (the environmental window which is an element in the standard burn plan used by the Agencies and TNC references in methods), and is mistakenly confused with the "prescription" in this study. Again, a simple attempt to relate the burn window to actual burn activities is suggested. Those records should be available from agency prescribed fire reports for the historical time of interest.

We greatly appreciate the reviewer's insights here. The authors agree that our word choice in the original manuscript was not consistent throughout the document, and have harmonized our terminology throughout the revised manuscript to emphasize the following: the prescription, which includes various environmental parameters considered acceptable during the burn, is not the only consideration with respect to the overall burn plan.

In the introduction, we have defined the prescription as the defined range of weather and live and dead fuel moisture conditions that enable managers to meet objectives while also minimizing the potential for control difficulties. We then define a "burn window" as time periods where a given location meets that prescription.

We have added the following text to make this explicitly clear:

"Historically, most prescribed burning in the WUS has occurred during spring or autumn, when weather and vegetation conditions are more likely to result in fire behavior that meets objectives but can still be controlled. These conditions collectively make up the "burn prescription" – or specified ranges of weather and live and dead vegetation moisture parameters that ensure fires burn completely enough to achieve objectives (e.g., consuming large fuels and woody debris) but that they don't burn so hot as to present control problems or to have undesired ecological consequences (e.g., increased tree mortality). Periods of time where these prescribed conditions are likely to be met are called the "burn windows." (Lines 76-83)

As discussed above, and alluded to by the Reviewer in the previous comment, comparing actual prescribed fires conducted to theoretical burn windows would be a labor-intensive task beyond the scope of the analysis described in this work. This would involve tracking down prescribed fire reports from the many agencies, units and districts implementing prescribed fire for the entirety of our historical period. In many cases, such data appears simply to not be available or to have been archived by the various entities that have implemented burns. Instead, we develop

environmental windows based on samples of burn plans from a range of geographies, ecosystems, and objectives to the best of our ability.

In addition, we emphasize that we are modeling weather and climate across broad regions for the purpose of comparing similarly broad regions and looking at potential long-term trends, with the understanding that such a process cannot and should not be used in place of an site-specific assessment of conditions on the ground (i.e., we simply cannot adequately simulate microclimates at the spatial scale that would be required to obtain real-world, site-specific details relevant to a specific actual or proposed prescribed fire). To do so, even if the data were readily available, would risk an “apples to oranges” comparison that might be misleading.

We certainly agree that the two referenced papers (Bajjnath-Rodino et al. 2022, Kupfer et al. 2020) offer excellent sub-regional to regional perspectives on this issue. We already cited both papers in our original manuscript, but now do so in additional locations to acknowledge their contributions more prominently and to emphasize that our analysis is on a considerably wider spatial scale.

6) Lastly, critical details must be added to the methods on the burn plans used: How many burn plans were used to develop the prescription window and produce a map of their geographic coverage in S1. This reviewer could not find this information and suggests that actual burn plans used also be archived with supplemental information (currently unavailable to this reviewer to assess the geographic distribution or relevance of burn prescription details across such a broad region). Given the broad regional differences in burn plan description of burn windows and objectives, the lack of subregional criteria is problematic to the overall analysis.

We greatly appreciate the reviewer raising this point—and we strongly agree. We apologize that details regarding geographic distribution and environmental envelopes from the specific real-world burn plans used to define RxDays were not available to the reviewer at the time of the original review. We have now fully rectified this problem by uploading the underlying environmental windows contained within each specific burn plan used in this study to a public repository (a direct link is available in the revised manuscript in the “Data Availability” section of the acknowledgements, and via this URL: <https://doi.org/10.5281/zenodo.7864935>).

Additionally, a table of values from these burn plans used in the analysis can be found directly in the Supplementary Materials as Table S1.

As the reviewer suggests, we have generated a new figure (Figure S1) that depicts the approximate location of each specific burn plan used in the study on a map of the WUS study domain (denoted by stars). To aid in contextualization of these burn plans, we also shade all

regions classified as “primarily forested” by LANDSAT data (details described below)--together illustrating both the overall geographic distribution of the plans used and the general vegetation type present in those locations. The figure has been reproduced below for convenience:

We have also, as noted above, implemented a substantial change to the underlying analysis in response to this feedback from the reviewer: we now use spatially-varying RxDay definitions according to coarse-scale predominant vegetation type (forest versus non-forest), using only the subset of available burn plans drawn from “primarily forested” versus “primarily non-forested” landscapes to create these underlying definitions. Although we do not find that these changes to the methodology change our overall findings, we feel that this approach is indeed more rigorous and does yield some modest, but interesting, sub-regional shifts in the projected spatial patterns of RxDay trends over decades (as is now discussed in the revised Results and Discussion sections).

In the Methods section, we now explicitly include information related to which ones are for “forest” versus “non-forest” ecosystems:

“These data were further separated into burn plans associated with predominantly forested landscapes and those associated all with other landscapes (i.e., “non-forest” vegetation types)...Separate thresholds were developed for forest and non-forest burn plans. Multiple environmental parameters exhibit modestly different ranges in forest versus non-forest plans (ST1), but the most notable difference is the absence of 100-hr and 1000-hr dead fuel moisture criteria in all non-forest plans (so such criteria are not included in our “non-forest” RxDay definitions). We apply these thresholds on a spatially explicit (grid box) basis, applying “forest” RxDay definitions on a spatially varying basis in locations classified as “forest” or “woodlands” by the Environmental Site Potential (ESP) product of LANDFIRE⁵² as in previous work⁹ and applying “non-forest” RxDay definitions in all other locations.” (Lines 423-437)

Finally, the other requested information about burn plans including the overall number of plans and the predominant coarse-scale vegetation type in which they were implemented is now explicitly included in brand new section in the Methods:

“Additional information regarding specific burn plans used in this analysis

We compile a total of 22 burn plans from across the WUS domain for use in this analysis. 14 of these plans are from primarily forested landscapes, and 8 are from primarily non-forested landscapes. The geographic distribution of the plans extends widely across the study domain, including 12 plans from California, 2 each from Colorado, Idaho, and Nevada, and 1 each from Washington, Oregon, Wyoming, and New Mexico (Fig. S1). The ranges of environmental conditions contained in forest versus non-forest plans is included in Table S1, and the specific parameters used in each plan may be found in the public repository noted in the Data Availability section.” (Lines 482-490)

Minor Comments:

Abstract

Line 26: “Purposeful burning” is not an appropriate definition of prescribed fire in this context, and the description that it is “challenging” to implement is only regional. Arson is purposeful.

We thank the reviewer for this insightful comment, which has spurred us to change the language in the manuscript in multiple locations to reflect this reality more accurately and consistently. We agree that “purposeful burning” was imprecise, and have made the following changes in various sections of the manuscript accordingly:

The first such instance is in the abstract, which now includes a more specific definition involving intention ignition specifically to achieve certain beneficial outcomes, now reads:

“However, the intentional use of fire as a vegetation management tool—known as “prescribed fire”—can reduce the risk of destructive fires and restore ecosystem resilience.” (Lines 31-33)

Then in the introduction, where the text now reads:

“Prominent among proposed potential strategies in addressing the WUS wildfire challenge is the use of prescribed fire (also known as controlled burning), which is the practice of intentionally igniting and managing fire under prescribed conditions to meet specific desired hazard reduction or ecosystem-related objectives—including the reduction of wildland fuel density and improving ecosystem health and resilience to a warming climate and other disturbances.” (Lines 60-65)

Additionally, the revised text in the discussion now reads:

“Increased fire on the landscape is inevitable in a warming climate, but there are active choices to be made regarding whether that fire comes in the form of increasingly high-intensity conflagrations—due to further increases in fuel loading, long-term vegetation aridification, and increasingly extreme fire weather conditions—or as the result of carefully planned, generally lower-intensity, and typically net-beneficial prescribed burning.” (Lines 331-336)

And:

“As has been evidenced by public policy conversations and real-world fire suppression decisions made in response to high-profile prescribed fire escapes^{36,37}, contemporary practitioners of prescribed fire are operating under climate and vegetation conditions that are increasingly outside the envelope of historical conditions³².” (Lines 352-356)

Finally:

“Ultimately, implementation of management, funding, and regulatory policies that enable greater flexibility by all kinds of “intentional burners” (i.e., prescribed fire and cultural burning practitioners collectively) would represent a climate adaptation commensurate with the projected shifts in WUS prescribed fire windows we find in this analysis.” (Lines 397-400)

The abstract while beginning with a focus on the west loses that regional context quickly, which should be clarified in summary of results since these findings mirror other recent studies on the topic.

We thank the reviewer for this feedback. Although we are subject to severe space constraints (i.e., a hard limit of 150 words) in the abstract, we have added as many mentions to the “Western United States” (abbreviated as “WUS” in the abstract following the first usage) as possible in the summary of results at the end.

The revised portion of the abstract now reads:

“Here, we quantify observed and projected trends in the frequency and seasonality of WUS RxDays. We find that while ~2C of global warming by 2060 will reduce overall WUS RxDays (-17%), particularly during spring (-25%) and summer (-31%), winter (+4%) may increasingly emerge as a comparatively favorable window for prescribed fire especially in northern states.” (Lines 35-39)

Introduction

Line 36: With wildfire crisis in quotes, there should be a citation, presumably the USFS report of that title.

We have removed the quotes here, since our original reason for including this language in the introduction was not only the USFS report but also other recent published reports as well as common popular usage of the term to describe recent trends in wildfire (and, in particular, wildfire-related losses) in the American West.

Line 54: use a standard definition of prescribed fire. Restoring precolonization processes is not an ecologically sound objective for prescribed burning nor language commonly used for current burn plans or ecological objectives. Increasingly, resilience is the objective in light of invasive species and climate change. restoring to past processes is inconsistent with the modeling results.

We agree, and have modified/standardized the text to reflect precisely this. The revised text in the introduction now reads:

“Prominent among proposed potential strategies in addressing the WUS wildfire challenge is the use of prescribed fire (also known as controlled burning), which is the practice of intentionally igniting and managing fire under prescribed conditions to meet specific desired hazard reduction or ecosystem-related objectives—including the reduction of wildland fuel density and improving ecosystem health and resilience to a warming climate and other disturbances.” (Lines 60-65)

Line 69: It seems odd to present summary results here, and there is repetition from the Abstract.

We appreciate the reviewer's concern here. However, to our knowledge, it is *Nature* family journal style that a short summary of overall results appear at the end of a relatively brief introduction in the "here, we show..." form. Additionally, we do feel it is important to be as specific as possible regarding results in the Abstract for clarity of communication—so it is true that a brief statement of our topline findings do appear both here and in the abstract. If the Editor this is out of place, we could certainly consider alternatives.

Line 74-75: This statement on 1000-hr fuels is incorrect and must be removed. If a focus of the paper, it must be given attention in the introduction as there is considerable literature on the topic. The importance of this fuels in escape risks has been recognized for decades in fire management both in wildfire and prescribed fire containment. Considerations of smoldering fuels is implied in burn plan reliance on KBDI and other Drought indices that also incorporate other smoldering fuel potential, including organic soil horizons (e.g., duff). See problems with analysis of 1000-hr fuels regionally in Figure comments.

We thank the reviewer for this perspective. It is true that 1,000 hour fuels have been indirectly considered via various indices, but Kolden (2005; later partially published in Kolden and Brown 2010) found that prescribed fire managers generally did NOT utilize KBDI and other indices that track large fuel moisture and climatic influences when making prescribed fire decisions, and this was particularly true in the WUS (where KBDI is not widely utilized as compared to the eastern states). While the most recent version of the *National Standards for Prescribed Fire Planning and Implementation* publication (PMS 484) includes a Lessons Learned box describing the utility of using such indices to reduce potential of an escape, the absence of both 1,000-hr fuel moisture prescriptions and any mention of drought or indices that track drought impacts on large fuels in nearly all of our burn plans suggests that this knowledge has been slow to trickle down into the planning process. This is potentially because most burn plans are based off of prior burn plans for a unit (so as to learn from predecessors), not written from scratch, and the utility of Fire Behavior calculators such as Behave does not require 1,000 hr FM to produce basic fire behavior.

We acknowledge that "jackpots" of heavy fuels on the fire line and overall fuel moistures have been recognized for decades in terms of escape risk, but this risk has not been translated into prescriptions that include 1000-hr FM. Large fuels have mostly been a concern for smoldering combustion historically (thus their inclusion in FOFEM), not for contributing to extreme fire behavior during the shoulder seasons. We hope our added texts conveys this more clearly.

As we note in the text, not all burn plans include prescription parameters for 1000 hour FM, because 1,000-hr fuel moisture is not required for burn plans. In addition, one of the core findings from the Gallinas prescribed fire review (of the Hermit's Peak and Calf Canyon fires) and USFS National Review, is that long-term drought indices (the report names the potential to use Evaporative Demand Drought Index) need to be considered, precisely because they haven't explicitly been included in prescriptions. In addition to drought indices, NFDRS indices such as ERC could also be informative as ERC incorporates large diameter fuel moisture. The burn plans we reviewed did not explicitly consider these indices either.

We have added a passage to the main text of the manuscript to articulate this reality in the Western United States:

“Whereas temperature, relative humidity, wind, and small-diameter fuel moisture are included in every burn plan reviewed for this study, and consistently encountered by the authors in the field, 1000 hour FMs are not consistently included in burn plans in the WUS. Yet our work shows that their inclusion has a large impact on potential RxDays in forested ecosystems. Critically, we find nearly ubiquitous projected decreases in 1000-hr FM across all seasons and in nearly all regions, with the greatest decreases generally occurring in MAM and JJA (Fig. 5). (We note, however, that such decreases are only relevant in areas where heavy dead and down fuels are actually present; this generally excludes grassland and sagebrush-dominated ecosystems except in localized forested “fuel jackpots.”) These widespread decreases in large-diameter dead fuel moisture, and their contribution to undesirable fire effects (including difficult-to-control fire behavior and enhanced smoke production and carbon emissions³⁶), highlight the importance of explicitly including metrics such as 100/1000-hr FM and/or drought indices in burn prescriptions to better reflect long-term aridification trends amid a warming climate.” (Lines 289-301)

Rather than modeling the various indices and metrics that incorporate 1000 fuel moisture, instead we modeled it as a stand alone metric, particularly since that metric does already appear in some burn plans. In addition, though we are modeling 1000-hr FM, fire managers can actually collect physical samples to get locally specific 1000 hour FMs if they want, whereas indices will always require some modeled input.

We have also added some language later in the manuscript to better capture these nuances and clarify our intent, including the following new or revised section:

“Given recent high-profile public discussions regarding the implications of escaped prescribed fires occurring under conditions characterized by unusually low vegetation and dead fuel moisture³⁷, including specific calls to use indices relevant to large-diameter fuel moisture (e.g., Energy Release Component)^{36,38}, we quantify the importance of considering large-diameter dead

fuel moisture (FM) in WUS prescribed burn plans. To do so, we recalculate our historical RxDay baseline excluding 100hr and 1000hr dead fuel moisture constraints (that is, consideration of large diameter woody debris (~3cm to 20cm in diameter), which includes heavy dead brush as well as downed trees and logs, for comparison to our original RxDay definition (which includes these constraints).” (Lines 260-268)

Results

Line 216. One of the reports cited here relevant to coarse woody debris is from the 2000 Cerro Grande escape, further emphasizing the long-term realization of its importance in prescribed fire risk analysis (see above). There are additional studies of downed and dead fuels from across the west and their growing contribution to fire behavior. Furthermore, this study relies on modeled values here of putative drying dynamics of 1000-hr fuels with respect to their moisture dynamics. Those empirical data are from sound wood may not be relevant in an aridifying west with an abundance of decaying fuels from regional bark beetle die off a decade ago.

We agree with the reviewer that this is a complicated issue, and that our original manuscript did not fully explore this complexity. We have added an entirely new paragraph to the results to further discuss this issue, and to reference more recent work on the fire behavior dynamics associated with very dry 1000-hr fuel conditions (Stephens et al. 2018 and 2022):

“Historically, 1000-hr fuel moisture has not been regularly considered in many burn prescriptions for forested areas in the WUS (of the 14 burn plans implemented in predominantly forested areas used in this study, only one includes a 1000-hr fuel moisture constraint) because it was a minimal component of broadcast burns where objectives were to consume smaller fuels and large fuel moisture was generally too high during burn windows for such fuels to combust. However, with warming temperatures this historical assumption may no longer be valid; in addition, the consideration of large diameter fuel moisture can be used as a proxy for longer-term trends in drought and aridity. Specifically, recent reviews have called for the use of drought indices (such as the Evaporative Demand Drought Index) from NFDRS National Fire Danger Rating System (NFDRS) indices in burn prescriptions^{36,38}, some of which consider large-diameter fuel moisture (e.g., Energy Release Component from NFDRS). Recent work has also suggested that larger-diameter fuels can contribute to “mass fire” events^{41,46}, suggesting they have a greater influence on fire behavior than previously believed and underscoring the importance of their explicit consideration.” (Lines 269-282)

We also note that the Cerro Grande report actually does not explicitly call out coarse woody debris, per se—instead, it simply acknowledges very dry fuels in general and the presence of jackpots on the fire line. However, as discussed above, the Hermit’s Peak fire review and the

National Review do specifically propose consideration of NFRDS indices, which offers direct evidence that this is indeed something that is only being considered relatively recently.

Figure 5. The inclusion of extensive sagebrush and short grass prairie in the 1000-hr dead fuel moisture calculation is inappropriate. Those systems (e.g., eastern MT, large portions of WY, and eastern NM, do not support forested vegetation relevant to this graphic, which is very misleading as those areas are shown to have the greatest potential change in 1000-hr fuel moisture.

We thank the reviewer for this feedback. In response to this and other concerns from both Reviewer #1 and Reviewer #2, we now use two separate RxDay definitions (i.e., “forest” and “non-forest” parameters) to better differentiate between conditions drawn from burn plans in different broadly defined vegetation classes. The new definition for non-forest RxDays, which encompasses most of the grasslands along the Rocky Mountain Front Range and adjacent Plains, no longer includes 1000-hr FM. These findings are reflected in the revised RxDay statistics, and associated figures showing these data, as described above in the Response Document and in the revised manuscript.

For uniformity, we do still include a domain-wide assessment of 1000-hr FM changes in Figure 5 as it is still relevant across a majority of the study domain. (The authors also note that even within broader prairie grasslands, there are still substantial pockets of trees and forest that can act as fuel jackpots responsive to 1000-hr fuel conditions—a reality that one study author personally witnessed during Colorado’s destructive Marshall Fire in 2021). However, we agree with the reviewer that this figure may well have been confusing as presented in the original manuscript absent this context. To address this, we have modified both the associated discussion in the text as well as the relevant future caption itself to emphasize that the values on this map are only relevant in a given location if there are actually heavy dead and down fuels present (which is often not the case in prairie/grassland/sagebrush ecosystems).

The revised text now reads:

“We note, however, that such decreases are only relevant in areas where heavy dead and down fuels are actually present; this generally excludes grassland and sagebrush-dominated ecosystems except in localized forested “fuel jackpots.” “ (Lines 294-297)

And additional text added to the Figure 5 caption reads:

“Note: these values are only relevant in places where heavy dead and down fuels actually exist locally.” (Line 702)

Discussion

Line 272: The use of “purposeful” should be replaced by prescribed fire (see comment above). If trying to include the expansion of managed wildfires for objectives, be consistent with policy terms in the region rather than the vague term “purposeful.”

We thank the reviewer for this insightful comment, which has spurred us to change the language in the manuscript in multiple locations to reflect this reality more accurately and consistently. Please see our response to the reviewer’s related comment regarding Line 26 for a full enumeration of all relevant text changes.

Methods

It must be mentioned that time lag fuels used in prescribed fire planning and NFDRS are constructs used primarily for wildfire behavior under extreme conditions. The metrics while common have little to no broad association with prescribed fire planning and thus drought indices are more commonly used as a surrogate (but only included in prescriptions through burn planner experience or regional rules of thumb). In the eastern US and portions of CA, for example, the range of acceptable 1000-hr fuel moistures in S-1 (10-20%) would represent unacceptably significant risk of smoldering consumption.

As we note above in response to the question about 1,000-hr fuels and the use of drought indices, our goal here was to highlight that large fuels should be considered because many recent escapes note the role of large fuels in contributing to the escape, but that historically large fuels have not been explicitly considered as part of the burn plan prescription. We recognize that smoldering combustion has been the primary concern related to large fuels historically (which is why they are included in FOFEM), which is consistent with our added description in the text of why large fuels have not historically been considered in the prescription (because they were largely too wet to burn). We hope that the considerable amount of text we have added in the Introduction and Methods helps to clarify why we are considering 1,000 FM here, and how the role of large fuels in prescribed fire behavior and outcomes has changed over time.

One such section now reads:

“Beyond direct meteorological constraints, climate change has already demonstrated indirect impacts on prescribed fire implementation. Extreme meteorological events, especially severe to historically unprecedented drought condition, have been contributing factors to several prescribed fires that “escaped” and became disastrous wildfires. The societal and political fallout from such events has led to multiple temporary U.S.-wide moratoria on prescribed fire (such as after the 2000 Cerro Grande Fire in New Mexico³⁷ and the Hermits Peak Fire in 2022³⁸). Longer fire seasons also mean that fire personnel are committed to fighting fire in some regions—which in turn reduces the number of trained personnel available to implement burns in other regions where conditions are favorable. In addition, longer fire seasons are particularly

challenging for federal agencies since their firefighting workforce is dominated by seasonal employees who can only work for limited durations³³⁻³⁵).” (Lines 93-103)

In addition to the more extensive new text added to the Introduction addressing this (copied above and earlier in this response document), we also added clarifying text to the Methods: *“Although NFDRS is generally used by fire managers to track broader trends in seasonal wildfire risk, we use it to derive these specific variables because they are also often included in burn prescriptions.”* (Lines 461-463)

References

Is “Fire, C.” the proper citation?

Thank you for catching this typo, which we have corrected. This reference should read: “Cal Fire.”

Reviewer #2 Remarks:

General comments:

The manuscript is well written and addresses a very important problem - i.e., how Rx burn windows might change in the western US. I have some general broad-level questions and concerns. Therefore I recommend major revisions.

We thank the reviewer for their acknowledgement of the importance of the issue at hand, and appreciate that they feel the original manuscript is well written. In response to the reviewer’s thoughtful feedback, we have conducted both substantial new analysis as well as made multiple changes to the text to reframe and clarify certain elements. A portion the revisions made in response to Reviewer #1 are also relevant in responding to Reviewer #1’s concerns, so we have discussed changes made in both sections of this document to make it as simple as possible for both reviewers to assess our revisions.

How different would the analysis be if you used local prescriptions for what constitutes one Rxdays? I would recommend conducting an uncertainty analysis if you used spatially different burn prescriptions.

This is an excellent suggestion, and we have directly implemented it. We now use spatially-varying definitions of RxDays across the WUS on a “macroecology” basis—that is, we make a simplified distinction between regions with “forest” versus “non-forest” predominant vegetation types. The revised manuscript reflects this modified analytical approach.

Additionally, we now also conduct additional analysis to quantify the degree to which these

different definitions affect our results on a domain-wide basis (i.e., the uncertainty associated with the spatially varying RxDay definition). In doing so, we now report that while the *baseline* number of RxDays is sensitive to the use of forest versus non-forest RxDay definitions (as expected due to the exclusion of FM100/1000 from all non-forest burn plans), long-term *trends* in RxDays (which is the focus of our analysis) are *not* highly sensitive to the specific RxDay definition used (historical trends change by <3% when applying forest versus nonforest RxDay definitions across the entire domain).

We have generated new figures S8 and S9, which show these results across the WUS domain, and have added a new sections to the Methods to discuss this sensitivity/uncertainty analysis:

“Testing sensitivity of baseline and trends to underlying RxDay definitions

As noted above, we use large-scale vegetation type-aware definitions for RxDays (i.e., forest versus non-forest regimes) that vary across the landscape according to the predominant vegetation type in a particular 4x4km grid box. To systematically test the sensitivity of both RxDay baselines and historical trends, we compare three possible RxDay definitions: first, one in which all available burn plans (covering forest and non-forested regions) are used to define acceptable ranges for environmental parameters; second, one in which only those burn plans from forested landscapes are used; and third, one in which only those burn plans from non-forested landscapes are used. We then apply these definitions uniformly across the landscape, irrespective of the actual vegetation type in a particular location, to explore the widest plausible parameter space from among our sample of real-world burn plans. Specific differences between the environmental conditions using each of these definitions can be found in ST1, though the most notable difference is that FM100/1000 are not included in non-forest plans.

We find that there is high sensitivity to the use of forest versus non-forest plans with respect to annual RxDay baseline occurrence (around 66 RxDays per year on a domain-wide average using non-forest definitions uniformly across the landscape, and 24 RxDays per year applying forest definitions uniformly across the landscape; Fig S8), which is expected given the complete exclusion of large-diameter fuel moistures from non-forest plans. However, we find low sensitivity to the use of different RxDay definitions to historical trends in annual RxDays on a domain-wide basis (a loss of 7 RxDays per year using a forest definition and a loss of 7.2 RxDays per year using a non-forest definition; also, note the spatially similar WUS trends in Fig. S9, with strong RxDay declines in the Pacific Southwest and Four Corners regions but little change or even slight increases across the Northern Tier). This indicates that key study findings are not strongly sensitive to RxDay definitions across the plausible range of values derived from real-world burn plans.” (Lines 492-516)

For convenience, we reproduce new Figure S9 below:

Fig. S9. Maps depicting sensitivity of projected RxDay trends to different RxDay definitions. Maps depicting the difference in protected trends in RxDays across the western United States

(WUS) on an annual basis depending on which specific RxDay parameter ranges are used. **a)** Change in annual RxDays using the median environmental condition parameters from all available burn plans. **b)** Change in annual RxDays using the median environmental condition parameters only from the subset of burn plans conducted in primarily forested settings. **c)** Change in annual RxDays using the median environmental condition parameters only from the subset of burn plans conducted in primarily non-forested settings. All RxDays are calculated using observed meteorological data from the MACA dataset over the years (1981-2060).

I am not fully sure about the LLS analysis. It is of course dependent on other factors such as wind, terrain, and local microclimatic situations. How well is it captured and how is it explicitly considered in the burning window prescription?

Thank you for raising this issue. We first note that the LLS metric is designed to capture conditions that affect ventilation on regional scales, independent of local topographic effects or microclimates. It is not explicitly considered in burn window prescriptions by practitioners on the ground; rather, it is meant to be a companion analysis that demonstrates future increases in air stagnation which could impact RxFire implementation via impacts to air quality. We emphasize that the LLS-related results do not affect the results in any other section of the paper, as we do not explicitly include an air stagnation element in the RxDay metric (due to widely varying regulatory guidelines across the study domain). We have added text in two places to the methods section of the manuscript to clarify our usage of this metric accordingly:

“Although this metric does not account for local topographic effects on smoke dispersion, it broadly captures conditions that would inhibit ventilation at regional scales.” (Lines 553-555)

and:

“We also note that air quality indicators are used in widely varying ways in different regions and under different administrative jurisdictions to manage particulate matter pollution risk, so

our use of the LLS in this context is an indirect proxy for air quality constraints rather than an explicit representation of regulatory constraints on prescribed fire.” (Lines 559-563)

How reliable are the estimates for 1,10,100 and 1000 hr fuel moisture?

We calculate NFDRS indices following the (extensively documented) operational methods, which rely on a series of equations and daily weather information. The development of NFDRS did attempt to relate derived 1- 10- 100- and 1000-hr dead fuel moisture to experimental samples. In our analysis, the major source of uncertainty in our estimates comes from the gridded nature of surface meteorological variables in gridMET—which can be particularly pronounced in complex terrain where microclimates are present. However, the operational approach for obtaining NFDRS indices often uses the ‘nearest’ automated weather station which may be located tens of kilometers away from the burn site and in different topographic siting. For these reasons, we do not expect the method of estimation used herein to be a primary source of error in our results.

In the revised methods section, we clarify this by noting:

“To calculate NFDRS metrics (ignition component, 1-hr, 10-hr, 100-hr, and 1000-hr dead fuel moisture), we primarily use traditional formulas originally described by⁵³. For all NFDRS metrics calculated using downscaled climate model ensemble (CMIP5/MACA) data, we apply a secondary bias correction to account for biases arising from covariance and serial correlation in the underlying data (e.g., ⁵⁷) using a quantile mapping approach (i.e., comparing downscaled MACA data to the original gridded metrics from gridMET pseudo-observations).” (Lines 540-545)

And by referencing:

Cohen, J. D. & Deeming, J. E. The national fire-danger rating system: basic equations. (U.S. Department of Agriculture, Forest Service, Pacific Southwest Forest and Range Experiment Station, 1985).

I think more information is needed to understand how to estimate the optimum Rx window. Can you provide the different Rx burn prescriptions used in the study? Which prescriptions are used in which region? How do you know that the prescription you derive is optimal?

The reviewer’s points here are well taken, and align with additional feedback from Reviewer #1.

We have made multiple extensive revisions to the text and conducted new analysis to address these concerns.

First, we have removed all language in the manuscript surrounding “optimality” for prescribed fire. We agree this was a potentially misleading use of the word, since what we are really assessing is whether meteorological and vegetation moisture conditions are within “acceptable bounds” (which may or may not overlap with what is truly “optimal”).

Next, to further address this concern, we have added the following text to the manuscript in addition to removing the “optimality” language:

“In this analysis, the quantitative framework we use to define a composite prescribed fire metric (i.e., to determine whether conditions, in aggregate, are within acceptable bounds for prescribed fire in a given location on a given day) is based upon a suite of near-surface meteorological variables (including air temperature, relative humidity, and wind speed)...” (Lines 456-459)

And:

“An important caveat regarding our composite RxDay definitions is that this bulk metric cannot be, and is not intended to be, fully representative of highly localized microclimate variations—nor does it capture various ecological, legal, and/or regulatory constraints that can and do dictate on-the-ground decisions regarding whether to proceed with a particular treatment on a given day. As such, the RxDay values and trends discussed in this analysis cannot be directly compared to real-world implementation of prescribed fire in specific locations. Any real-world treatment would require detailed local weather and vegetation information that is simply not available even at the horizontal resolution (i.e., $\sim 16\text{km}^2$) of the downscaled climate model data used in this study. Instead, our goal here is to assess broad-scale trends in meteorological conditions relevant to prescribed fire across the WUS while acknowledging that local definitions and realities may differ considerably and in consequential ways on specific prescribed burns.” (Lines 470-480)

Additionally, we previously used only a single RxDay definition across the entire domain. We have now revised the entire analysis to use two different definitions: one drawn from real-world burn plans in “non-forest” regimes, and another from real-world burn plans in “forest” regimes. We apply these different definitions in a spatially explicit manner across the landscape. While we acknowledge this still does not encompass the full range of ecosystem and vegetation type complexity across the broad study domain, we feel that this does capture the key differences between different macroregions. (Results from these revised analyses are described both in prior sections of the Response Document as well as in the revised manuscript itself).

Finally, we apologize that the specific real-world burn plans used to define forest and non-forest RxDays were not available to the reviewer at the time of the original review. We have now

rectified this problem by uploading the underlying burn plan weather and environmental data to a public repository (a direct link is available in the revised manuscript in the “Data Availability” section of the acknowledgements.)

We have also updated Supplementary Table 1 to better reflect the new forest versus non-forest RxDay definitions, as well as the additional burn plans integrated into the analysis during the revision process. We have reproduced that table here as well for convenience.

Table ST1: Comprehensive list of prescribed fire criteria used in this study.

RxDay Constituent Variable	All Plans	Forest Plans	Non-Forest Plans
Number of burn plans	22	14	8
2m Air temperature (°C)	5.8-29.4	7.2-28.9	4.4-29.4
2m Relative humidity (%)	16-60	15-60	20-70
Scaled 10m (mid-flame) wind speed (m/s)	0-4.5	0-5.0	0-5.4
Ignition component (units)	20-65	20-75	21-60
1 hour fuel moisture (%)	4-12	4-12	4-13
10 hour fuel moisture (%)	6-14	6-14	6-15
100 hour fuel moisture (%)	8-15	7.5-15	–
1000 hour fuel moisture (%)	10-20	10-20	–

15th Aug 23

Dear Dr Swain,

Please accept my sincerest apologies for the long delay in obtaining reports on your revised manuscript titled "Climate change is narrowing and shifting prescribed fire windows in western United States". Your manuscript has now been seen by one of the original reviewers and a replacement reviewer (as Reviewer #1 was unavailable to comment on the revised manuscript). The latest reviewer comments appear below. In light of their advice we are delighted to say that we are happy, in principle, to publish a suitably revised version in Communications Earth & Environment under the open access CC BY license (Creative Commons Attribution v4.0 International License).

We therefore invite you to revise your paper one last time to address the remaining concerns of our reviewers. At the same time we ask that you edit your manuscript to comply with our format requirements and to maximise the accessibility and therefore the impact of your work.

EDITORIAL REQUESTS:

*****Please take care to match our formatting and policy requirements. We will check revised manuscript and return manuscripts that do not comply. Such requests will lead to delays. *****

SUBMISSION INFORMATION:

OPEN ACCESS:

Communications Earth & Environment is a fully open access journal. Articles are made freely accessible on publication under a [CC BY license](http://creativecommons.org/licenses/by/4.0) (Creative Commons Attribution 4.0 International License). This license allows maximum dissemination and re-use of open access materials and is preferred by many research funding bodies.

For further information about article processing charges, open access funding, and advice and support from Nature Research, please visit <https://www.nature.com/commsenv/article-processing-charges>

At acceptance, you will be provided with instructions for completing this CC BY license on behalf of all authors. This grants us the necessary permissions to publish your paper. Additionally, you will be asked to declare that all required third party permissions have been obtained, and to provide billing information in order to pay the article-processing charge (APC).

[link redacted]

Best regards,

Clare Davis, PhD
Senior Editor
Communications Earth & Environment

www.nature.com/commsenv/
@CommsEarth

REVIEWERS' COMMENTS:

Reviewer #2 (Remarks to the Author):

Thanks for addressing the comments and I am satisfied with the revisions. Especially, clarifying some of the languages regarding 'optimality' and the spatially varying analysis along with the sensitivity studies clarifies the manuscript significantly. I recommend publication in the current form.

Reviewer #3 (Remarks to the Author):

Major comment 1:

Authors have sufficiently addressed reviewer comment.

Major comment 2:

Authors have sufficiently addressed reviewer comment.

Major comment 3:

Authors have sufficiently addressed reviewer comment.

Major comment 4:

In this reviewer's opinion, the information under subheading "Additional information regarding specific burn plans used in this analysis" should be merged with what is now the second paragraph

of the methods beginning page 18. The additional section is repetitive, while the original paragraph lacks necessary detail. With this minor change, the author will have fully addressed reviewer comment.

After writing the above, I see that the subheading “Additional information regarding specific burn plans used in this analysis” is new and in response to major comment 6. Please still consider the suggested change, though it constitutes only a minor comment.

Major comment 5:

Authors have sufficiently addressed reviewer comment.

Major comment 6:

See Major comment 4.

I suggest one or two additional sentences near the beginning of the methods providing justification that 22 burn plans are sufficient to represent appropriate burn windows for both forested and unforested landscapes across the WUS. How much do appropriate burn windows vary within regions? If there is low variability, 22 plans may be sufficient to capture regional differences across the WUS. If there is high variability, 22 plans may be few. Though this may be common knowledge in the industry, I believe it to be especially important for readers outside of the prescribed fire realm.

Otherwise, I believe the author has sufficiently addressed reviewer comment.

Minor comments

Minor comments were sufficiently addressed by the authors.

1. Use of “1000-hour” or “1000 hour” should be consistent (see pg 13).

Swain et al. 2023 Response to Reviewers

We thank the Editor and reviewers for their additional feedback in this second round of revision. We have slightly modified the manuscript in response to these modest comments and “accept in principle” editorial decision, which are enumerated in detail below.

This Response to Reviewers document provides a complete documentation of all changes made in response to the reviews. The document is designed so that the changes that we have made in response to each comment can be immediately read and understood, independent of the other comments and responses.

Reviewer comments are shown in **bold**. Author responses are shown in plain text. Passages excerpted from the main manuscript are *italicized*.

Reviewer #2 Remarks:

Thanks for addressing the comments and I am satisfied with the revisions. Especially, clarifying some of the languages regarding 'optimality' and the spatially varying analysis along with the sensitivity studies clarifies the manuscript significantly. I recommend publication in the current form.

We thank the reviewer for assessing the revised manuscript. We are that glad the completed revisions have satisfactorily addressed the reviewer’s previous comments.

Reviewer #3 Remarks:

We thank the reviewer for taking time to assess the revised version of this manuscript. We are pleased the new reviewer believes we have sufficiently addressed the major comments from the previous reviewer, and respond to the remaining minor comments below.

Major comment 1:

Authors have sufficiently addressed reviewer comment.

Major comment 2:

Authors have sufficiently addressed reviewer comment.

Major comment 3:

Authors have sufficiently addressed reviewer comment.

Major comment 4:

In this reviewer’s opinion, the information under subheading “Additional information regarding specific burn plans used in this analysis” should be merged with what is now the second paragraph of the methods beginning page 18. The additional section is repetitive, while the original paragraph lacks necessary detail. With this minor change, the author will have fully addressed reviewer comment.

After writing the above, I see that the subheading “Additional information regarding specific burn plans used in this analysis” is new and in response to major comment 6. Please still consider the suggested change, though it constitutes only a minor comment.

We thank the reviewer for this comment. As the reviewer notes, this section was added in direct response to a previous reviewer comment—and so despite the modest repetition this brings to the methods section, we would prefer to keep this section as it presently is given that all necessary information is now presented in this section.

**Major comment 5:
Authors have sufficiently addressed reviewer comment.**

**Major comment 6:
See Major comment 4.**

I suggest one or two additional sentences near the beginning of the methods providing justification that 22 burn plans are sufficient to represent appropriate burn windows for both forested and unforested landscapes across the WUS. How much do appropriate burn windows vary within regions? If there is low variability, 22 plans may be sufficient to capture regional differences across the WUS. If there is high variability, 22 plans may be few. Though this may be common knowledge in the industry, I believe it to be especially important for readers outside of the prescribed fire realm.

Otherwise, I believe the author has sufficiently addressed reviewer comment.

We thank the reviewer for this helpful feedback. As the reviewer hints, there is indeed low variability in burn plan parameters across the study region. In response, we have added the following paragraph to the section in question:

The modest number (n=22) of burn plans used in this study is justified due to the low variability in environmental parameters across burn plans (even across regions and ecotypes). This stems from the fact that most or all contemporary prescription windows use the same underlying

physical fire behavior models⁵². As biomass combusts and fires behave according to fundamental physical principles no matter the setting, the range of heat (in the form of temperature) and moisture (in the form of humidity and fuel moisture) requisite to achieve the desired level of combustion does not change across regions. Rather, our assessment of burn plans demonstrates that the timing and duration of such windows is the primary difference across biomes and regions—particularly when low-to-moderate intensity fire is the desired objective across most prescribed burns. (Lines 431-440)

⁵²Rothermel, R. C. (1972). *A mathematical model for predicting fire spread in wildland fuels (Vol. 115). Intermountain Forest & Range Experiment Station, Forest Service, US Department of Agriculture.*

Minor comments

Minor comments were sufficiently addressed by the authors.

1. Use of “1000-hour” or “1000 hour” should be consistent (see pg 13).

We suggest that journal “house style” will likely dictate the convention ultimately used for dashed quantity-units, so we defer to the Editor or copyediting team as to the appropriate format in such instances. For consistency, however, we have included the dash in all instances for the purposes of the resubmission.